# ALGORITHMIC DETERMINATION OF THE COMBINATORIAL STRUCTURE OF THE LINEAR REGIONS OF ReLU NEURAL NETWORKS

## ABSTRACT

We algorithmically determine the regions and facets of the canonical polyhedral complex, the universal object into which a ReLU network decomposes its input space. We show that the locations of the vertices of the canonical polyhedral complex along with their signs with respect to layer maps determine the full facet structure across all dimensions. Our algorithm which implements this approach makes use of our theorems that the dual complex to the canonical polyhedral complex is cubical, and it possesses a multiplication compatible with its facet structure. The resulting algorithm is numerically stable, polynomial time in the number of intermediate neurons, and obtains accurate information across all dimensions. This permits us to obtain, for example, the true topology of the decision boundaries of networks with low-dimensional inputs for binary classification tasks. We run empirics on such networks at initialization, finding that width alone does not increease observed topology, but width in the presence of depth does.

## 1 INTRODUCTION

For fully-connected ReLU networks (Nair & Hinton, 2010), the canonical polyhedral complex of the network, as defined by Grigsby & Lindsey (2020), encodes its decomposition of input space into linear regions and determines key structures such as the decision boundary for a binary classification task. Investigation of properties and characterizations of this decomposition of input space are ongoing, in particular with respect to counting the top-dimensional linear regions (Serra et al., 2018; Hanin & Rolnick, 2019a; Montufar et al., 2014; Serra & Ramalingam, 2020; Xiong et al., 2020), since these bounds give one measure of the expressivity of the associated network architecture. However, a theoretical understanding of adjacency between regions and more generally the connectivity of lower-dimensional facets are to our knowledge generally undocumented. Understanding the face relations in the canonical polyhedral complex–for example, for tiled surfaces, understanding all inclusions between vertices (0-faces), edges (1-faces) and polygons (2-faces) –is necessary to relate combinatorial properties of the polyhedral complex of a network to the topology of regions into which the decision boundary partitions input space, geometric measurements such as the presence of critical points (Grigsby et al., 2022), or other notions of topological expressivity, as explored by Guss & Salakhutdinov (2018); Bianchini & Scarselli (2014).

It is common to describe linear regions of the input space, $\mathbb{R}^{n_o}$, using "activation patterns" or "neural codes" recorded as vectors in $\{0, 1\}^N$ (Itskov et al., 2020). Unfortunately, having a list of which activation patterns are present in the interiors of the linear regions does not determine their pairwise intersection properties (Theorem 15), and computing the intersections of these regions directly is not numerically stable. Furthermore, the polyhedra comprising the linear regions appear, at first glance, arbitrarily complicated. In this work, we establish a simpler representation by proving the theorem that the geometric dual[1] of any network's canonical polyhedral complex is a union of $n_0$-dimensional cubes (see Figure 1). Inspired by the theory of oriented matroids and hyperplane arrangements (Anders et al., 2000; Aguiar & Mahajan, 2017), we demonstrate how the notion of "sign vectors" from oriented matroids, which are vectors with entries in $\{-1, 0, 1\}$, can serve as a labeling scheme for vertices, edges, and higher-dimensional regions. These sign vectors have properties that track

---

[1]Recall, for example, that the dual of the icosahedron is the dodecahedron.

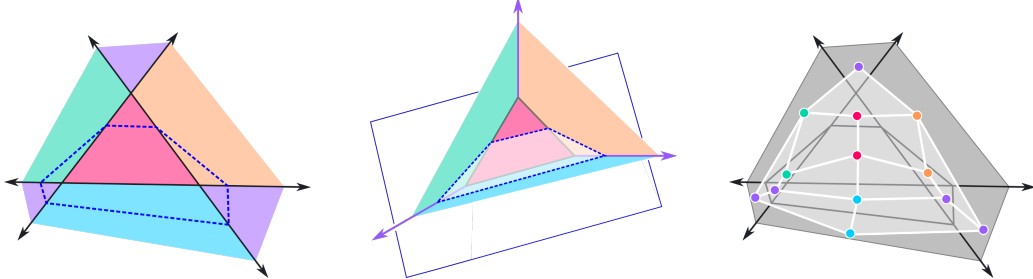

Figure 1: An illustration of a canonical polyhedral complex and its geometric dual. Left: $\mathcal{C}(F)$ for a specific neural network function $F : \mathbb{R}^2 \to \mathbb{R}^3 \to \mathbb{R}$. The three straight lines and the solid colored regions together form $R^{(1)}$, which is also $\mathcal{C}(F_1)$. Middle: $F_1 : \mathbb{R}^2 \to \mathbb{R}^3$ is piecewise linear on cells of $\mathcal{C}(F_1)$. The hyperplane in $\mathbb{R}^3$ is the hyperplane associated with $A_2 : \mathbb{R}^3 \to \mathbb{R}$, and this hyperplane together with the two halfspaces on either side of it form $R^{(2)}$. The cells of $\mathcal{C}(F)$ on the left are mathematically determined by taking one region $R$ of $R^{(2)}$, considering its preimage $F_1^{-1}(R)$, and taking the intersection of this preimage with a cell of $\mathcal{C}(F_1)$. Right: The geometric dual sign sequence complex $\mathcal{S}(F)$ is superimposed in white over $\mathcal{C}(F)$, with one vertex for each region of $\mathcal{C}(F)$. As we prove in general, $\mathcal{S}(F)$ is cubical, with each two-cube (quadrilateral) containing a unique vertex of $\mathcal{C}(F)$.

face relations and intersections between all linear regions in input space. We show that with full probability, computing only the vertices present in the polyhedral complex and recording their sign vectors gives enough information to determine all face relations in the polyhedral complex. The ability to compute the explicit decision boundary of a network provides a new means to evaluate topological expressivity of network functions.

Our main contributions are:

- Mathematically, we prove the existence of a combinatorial description of the geometric dual of the canonical polyhedral complex of a ReLU neural network which consists of a generalization of activation patterns, which we call the *sign sequence complex*. Furthermore, using a product structure adapted from the theory of oriented matroids which we prove to be well-defined, we show that the only information needed to determine the face poset structure of the sign sequence complex is the sign sequences corresponding to the vertices of the polyhedral complex.

- We implement an algorithm for obtaining these sign sequences which is numerically stable and, on average at initialization, runs in polynomial time in the total number of neurons, and exponential time in the input dimension.

- We show that the sign sequence complex can be naturally restricted to particular substructures of the polyhedral complex of a ReLU network. In particular, a chain complex computing the homology of the decision boundary can be obtained with simple operations on a subset of the vertices of the polyhedral complex, together with their sign sequences.

- We demonstrate the usefulness of this characterization of a network by obtaining the statistics of ReLU networks' binary decision boundaries' topological properties at initialization, dependent on architecture. These experiments provide empirics suggesting that depth of a network plays a stronger role in topological expressivity, at least at initialization, than the number of intermediate neurons.

In this work, our primary aim is to understand $\mathcal{C}(F)$ more completely. We do this by demonstrating how to use sign sequences to record all information about the adjacency of linear regions without needing to separately list all edges, faces, and higher-dimensional cells. Sign sequences have a geometric interpretation as a cubical complex, and an algebraic structure which is useful in proofs. It is also computationally tractable to treat these sign sequences as a *data structure* which provides a key for translating between *local* geometric and analytic information about $\mathcal{C}(F)$ (vertex locations and function behavior at those locations) and *global* topological and combinatorial information about the linear regions, including the previously-inaccessible topology of binary decision boundaries.

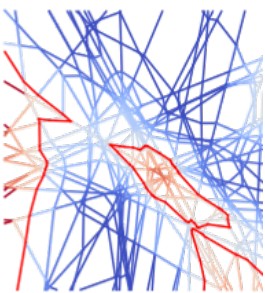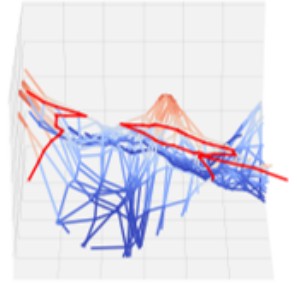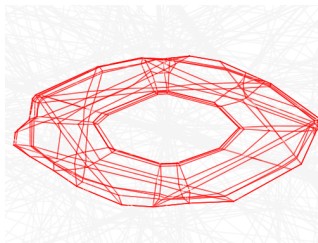

Figure 2: Left: $\mathcal{C}(\mathcal{F})$ for a a randomly-initialized neural network $F$ with architecture (2,15,15,1). Its decision boundary is in solid red and has nontrivial $\beta_1$. Middle: The graph of the function $F : \mathbb{R}^2 \to \mathbb{R}$, with the decision boundary as its zero level set. Right: In red, the decision boundary of a neural network with architecture $(3, 15, 8, 1)$. The network was trained using stochastic gradient descent to distinguish a points sampled from a torus from points sampled from an annulus at its center.

## 2 RELATED WORK

The seminal paper by Grigsby & Lindsey (2020) establishes a high-level view of the cellular structure of the canonical polyhedral complex, but does not establish explicit low-dimensional information. Under weak assumptions, they show the canonical polyhedral complex's $(n_0 - 1)$-skeleton (the set of $n_0 - 1$-dimensional polyhedra) may be described as the preimages of hyperplanes from each layer map, but arbitrary $k$-skeletons are unexamined for $k < (n_0 - 1)$, as are general face relations. The subsequent work (Grigsby et al., 2022) establishes local models for the polyhedral structure at the intersection of hyperplanes, but does not address deeper network structures as we do here.

Hyperplane arrangements are used by Hanin & Rolnick (2019a;b) to study linear regions, but primarily for computing volumes and counting top-dimensional regions, and not obtaining adjacency relations. In particular, in these works properties of hyperplane arrangements are used to establish *statistical* properties of the canonical polyhedral complex. While our work does rely on properties of hyperplane arrangements in a similar way, we focus on encoding the full face poset. In addition, others who approach explicit computation of linear regions as in (Zhang & Wu, 2020) do so using halfspace intersections, and we use vertices. While in theory one could intersect top-dimensional regions pairwise to obtain their shared faces, this is not numerically stable, especially when the linear equations involved arise from matrix multiplication. Our algorithm tracks known equalities and uses discrete signs to track face relations, avoiding issues potentially arising from numerical error in polyhedral intersection. Furthermore, in contrast to approaches by Curto et al. (2019) and other neuroscientific and biological applications where boundary structure is not clearly defined, in the context of artificial networks the boundary intersections are in fact computable.

Other characterizations of the combinatorics of ReLU networks' polyhedral complexes exist, but lack explicit implementation or applicability to deeper networks. Balestriero et al. (2019) describe the regions of the canonical polyhedral complex according to the roots of a polynomial, but no algorithm is presented on how to obtain these roots, nor how to explicitly determine whether two polyhedra are connected by a shared face. Zhang et al. (2018) provide a tropical characterization of the polyhedral complex including its face relations, but rely on the translation of network functions to tropical rational functions with integer coefficients, which is a discontinuous operation. In contrast, we conjecture that the signs of vertices present in the sign sequence cubical complex are stable in open sets of parameter space and could therefore be recorded to track changes in a network's topology during training. The subsequent application of tropical geometry by Alfarra et al. (2020) appears limited to networks with single hidden layers. Additionally, Itskov et al. (2020) establish a characterization of the regions of single-layer hyperplane networks which relies on similar sign labelings, but the methods do not apply to deeper networks.

Some reachability analysis algorithms using polyhedral methods such as those by Yang et al. (2020; 2021) use signs to determine how polyhedra have been "split" by a layer map, tracking the face lattices of the output polyhedra of each layer. In theory, this approach could plausibly record which

polytopes in the input space intersect in shared regions, but these papers consider polytopes independently from another in the next layer for purposes of parallelization. It is unclear whether enough information is retained to store which "splits" line up between polytopes, as this is not needed in their applications. Additionally, this approach requires storing all faces for the face lattice in each polytope, whereas we show this information is fully contained in the sign sequences of the vertices.

## 3 THE SIGN SEQUENCE CUBICAL COMPLEX

We define the sign sequence cubical complex and justify its importance before describing an algorithm for its computation in section 4.1. For readability, most terms are defined in prose, with fully precise definitions available in Appendix A.1.

### 3.1 PRELIMINARIES

We work with fully-connected, feedforward ReLU networks, following the framework by Grigsby & Lindsey (2020). In this framework, a network $F$ is expressed as $A_m \circ F_{m-1} \circ ... \circ F_2 \circ F_1$, where each $F_i = \text{ReLU} \circ A_i$ for an affine map $A_i : \mathbb{R}^{n_{i-1}} \to \mathbb{R}^{n_i}$, and the last affine map $A_m$ has image in $\mathbb{R}$ and is not followed by the ReLU function (Definition 3). We refer to the activity of each individual neuron of the network before the ReLU function is applied as the $(i, j)$th *node map*:

**Definition 5** ($F_{ij}$ Grigsby & Lindsey (2020), Definition 8.1 )**.** *If $F$ is a ReLU neural network, the* ***node map*** *$F_{ij}$ is defined by:*

$$\pi_j \circ A_i \circ F_{i-1} \circ ... \circ F_1 : \mathbb{R}^{n_0} \to \mathbb{R}$$

Note that $\pi_j$ is projection on to the $j$th coordinate, $i$ indexes layers, and $j$ indexes neurons.

For fixed $i$, the solutions to $\pi_j \circ A_i = 0$ are hyperplanes in $\mathbb{R}^{n_{i-1}}$, which together form a hyperplane arrangement (Aguiar & Mahajan, 2017). These hyperplane arrangements are equipped naturally with the structure of a polyhedral complex, which is an arrangement of vertices, edges, polygons, and generally $k$-dimensional polyhedra for $0 \leq k \leq n_{i-1}$ (Definition 1). We denote the polyhedral complex associated to $A_i$ by $R^{(i)}$ (Definition 3). The canonical polyhedral complex $\mathcal{C}(F)$ consists precisely of the polyhedra in $\mathbb{R}^{n_0}$ which may be described by selecting one polyhedron $R_i \subset \mathbb{R}^{n_i}$ from each hyperplane arrangement $R^{(i)}$, considering the preimages $(F_{i-1} \circ .. \circ F_0)^{-1}(R_i)$ in $\mathbb{R}^{n_0}$, and then taking the intersection of the resulting polyhedra (Definition 7, or see Figure 8). However, while this defines $\mathcal{C}(F)$ mathematically, this does not describe a tractable algorithm for its computation.

It is established by Grigsby & Lindsey (2020) and Grunert (2017) that each intermediate complex $\mathcal{C}(F_k \circ ... \circ F_1)$ is a polyhedral complex which subdivides the previous one, and resultingly $F$ is affine linear on each cell of $\mathcal{C}(F)$.

Our results apply to certain subsets of ReLU neural networks, called *generic* (Definition 10) and *supertransversal* (Definition 11), additional technical conditions which are nonrestrictive in light of the following lemma that almost all networks are supertransversal.

**Lemma 12.** *Supertransversality is full measure in $\mathbb{R}^P$, where $P$ is the set of network parameters. Additionally, it is fiberwise generic. That is, with fixed network weights, the set of biases corresponding to supertransversal networks is open and full measure.*

That almost all networks are, additionally, *generic*, was established by Grigsby & Lindsey (2020). As a result, in all but a measure-zero subset of networks, the theory developed below will hold.

### 3.2 THE SIGN SEQUENCE CUBICAL COMPLEX IS GEOMETRICALLY DUAL TO $\mathcal{C}(F)$

For networks with $N$ neurons, binary strings of length $N$ (which we will denote using -1 and 1) may serve as a labeling scheme to describe which neurons are active at a point in a ReLU network's input space (Itskov et al., 2020) on the interior of cells of $\mathcal{C}(F)$. However, this binary labeling is insufficient to encode all face relationships and the topology of a network's canonical polyhedral complex, which we prove in the appendix in Theorem 15. We propose instead to use *sign sequences*, defined below, as a means to label all polyhedral faces of $\mathcal{C}(F)$ and thereby fully encode the combinatorial properties of a network's canonical polyhedral complex.

**Definition 13.** *Define $s : \mathcal{C}(F) \to \{-1, 0, 1\}^N$ by $s_{ij}(C) = sign(F_{ij}(C^\circ))$. We call $s(C)$ the sign sequence of the cell $C$.*

Sign sequences are suffcent to list the cells of $\mathcal{C}(F)$. Though this is proven in various forms elsewhere (Jordan et al., 2019), we provide a different proof.

**Theorem 14.** *The function $s$ is well-defined and injective.*

Furthermore, the sign sequence of a cell is determinative under conditions of supertransversality and genericity. For example, it encodes the dimension of the cell.

**Lemma 16.** *Let $F$ be generic and supertransversal. Let $C$ be a $k$-cell of $\mathcal{C}(F)$. Then $s(C)$ has exactly $n_0 - k$ entries which are zero.*

This leads to a key *geometrically dual* relationship between the canonical polyhedral complex of a supertransversal network and a subcomplex of $[-1, 1]^N$, sending $k$-cells in $\mathcal{C}(F)$ to $(n_0 - k)$-faces of the cube. We will call this subcomplex $\mathcal{S}(F)$. Recall that a pure (sub)complex is a complex where every face is contained in some polytope of uniform top dimension.

**Theorem 20.** *For each generic, supertransversal neural network $F$, the image of the map $S : \mathcal{C}(F) \to \{-1, 0, 1\}^N$ gives a geometric duality between $\mathcal{C}(F)$ and a pure subcomplex $\mathcal{S}(F)$ of the hypercube $[-1, 1]^N$ endowed with the product CW structure.*

In particular, vertices of $\mathcal{C}(F)$ correspond to $n_0$-cells of $\mathcal{S}(F)$. Since the complex $\mathcal{S}(F)$ is pure and $n_0$-dimensional, knowing which $n_0$-cells are present is sufficient to determine all face relations in the subcomplex of the hypercube. A corollary is:

**Corollary.** *The sign sequences of the vertices of $\mathcal{C}(F)$ determine the face poset of the polyhedral complex $\mathcal{C}(F)$.*

### 3.3 A BINARY OPERATION ON SIGN SEQUENCES ENCODES THE FACE RELATIONS OF $\mathcal{C}(F)$

The sign sequence complex has only one combinatorial type, cubes. This uniformity is related to a package of formal properties which will be crucial in algorithmic implementation, with the existence of a multiplicative "composition" derived from oriented matroid theory (Anders et al., 2000) being particularly helpful.

**Lemma 18.** *If $C$ and $D$ are two polyhedra in $\mathcal{C}(F)$, the product $S(C) \cdot S(D)$ given by:*

$$(S(C) \cdot S(D))_{ij} = \begin{cases} S(C)_{ij} & \text{if } S(C)_{ij} \neq 0 \\ S(D)_{ij} & \text{otherwise} \end{cases}$$

*is well-defined as a binary operation between sign sequences of $\mathcal{C}(F)$. That is, there exists a cell $E$ in $\mathcal{C}(F)$ such that $S(C) \cdot S(D) = S(E)$ for all cells $C$ and $D$. Thus, sign sequences of $\mathcal{C}(F)$ form a semigroup.*

Following from similar constructions in hyperplane arrangements (Aguiar & Mahajan, 2017) we obtain the following:

**Lemma 19.** *For all supertransversal networks, the following relations hold for all $C$ and $D$ in $\mathcal{C}(F)$, where $\leq$ is the relation "is a face of":*

- $C \leq D$ *if and only if* $S(C) \cdot S(D) = S(D)$

- $S(C) \cdot S(D) = S(D) \cdot S(C)$ *if and only if there is a cell $E$ with $D \leq E$ and $C \leq E$.*

- $S(C) \cdot S(D) = S(C)$ *if and only if $C$ is contained in the intersection of the maximal set of bent hyperplanes which contain $D$.*

These characterizations are primarily useful for using code to track face relations via a discrete structure. As implemented in Section 4.2, we can use these to compute the topological properties of the decision boundary of a network using sign sequences. These properties can be seen illustrated in Figure 3.

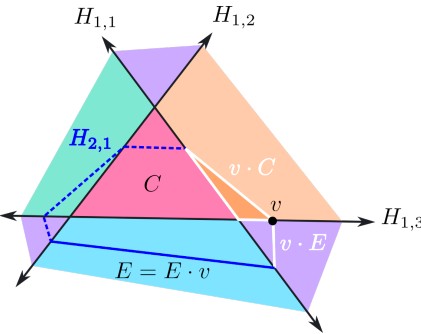

Figure 3: "Multiplication" of polyhedra on the sign sequence cubical complex, pictured geometrically. Under one possible co-orientation of hyperplanes, these cells have the sign sequences indicted in Table 1. The product is computed and pictured for certain pairs of cells.

Table 1: The sign sequence of the cells in Figure 3, together with some computed products.

| Cell | Sign Sequence |
|------|---------------|
| $v$ | (1,1, 0, 0) |
| $E$ | (1,1,-1, 0) |
| $C$ | (1,1, 1,-1) |
| $v \cdot C$ | (1,1, 1,-1) |
| $v \cdot E$ | (1,1,-1, 0) |

## 4 USING PROPERTIES OF $\mathcal{S}(F)$ TO COMPUTE $\mathcal{C}(F)$ AND ITS TOPOLOGY

The properties of the sign sequence complex make it straightforward to compute the combinatorial properties of the polyhedral complex of a network across all dimensions upon obtaining the sign sequences of the vertices of $\mathcal{C}(F)$. Moreover, these sign sequences follow from locating potential vertices, thus knowing locations of the 0 entries in its sign sequence, and then evaluating the network to obtain its remaining signs. We summarize the process here, and provide the locations in the appendix for more precise statements of each lemma.

### 4.1 OBTAINING THE VERTICES OF $\mathcal{C}(F)$ BY CONSIDERING SIGN SEQUENCES

Grigsby & Lindsey (2020) define the canonical polyhedral complex $\mathcal{C}(F)$ iteratively through layers. Letting $R^{(k)}$ be the polyhedral complex associated with the hyperplane arrangement in layer $k$, the complex $\mathcal{C}(F_k \circ ... \circ F_1)$ is given precisely by the intersection complex of $\mathcal{C}(F_{k-1} \circ ... \circ F_1)$ and $(F_{k-1} \circ ... \circ F_1)^{-1}(R^{(k)})$. (See Definition 7). To obtain the vertices of a particular network's canonical polyhedral complex, we may therefore begin by obtaining the vertices corresponding to $\mathcal{C}(F_1)$, its first layer's canonical polyhedral complex. The exact computations needed are provided in Lemma 22.

The sign sequences of the top-dimensional regions which are present in $\mathcal{C}(F_1)$ can be determined by the sign sequences of the vertices of $\mathcal{C}(F_1)$, ignoring signs corresponding to neurons in later layers. The exact coboundary operation defining how to obtain these sign sequences is described by Lemma 21.

Following the computation of the first layer, subsequent layers' vertices may be found by analyzing the preimage of each bent hyperplane for additional intersections of $k$ bent hyperplanes from the new layer together with $n_0 - k$ bent hyperplanes from the previous layers. Since $F_{k-1} \circ ... \circ F_1$ is affine on each region of $\mathcal{C}(F_{k-1} \circ ... \circ F_1)$, restricted to each region, $F_{ij}(x)$ is affine. For details on this recursive step, see Lemma 23.

We may therefore proceed iteratively through layers in order to obtain the full polyhedral complex. In summary,

**Computing Sign Sequences.** *To obtain the vertices of $\mathcal{C}(F)$, and thus the $n_0$-cells of $\mathcal{S}(F)$:*

1. *Compute the intersections of the hyperplanes from the first layer, as in Lemma 22. Obtain their sign sequences by evaluating $F_{ij}$ on each intersection. Restricting these sign sequences to the signs of $F_{1j}$ obtains $\mathcal{C}(F_1)$.*

2. *To compute $\mathcal{C}(F_i)$, loop through regions $C$ in $\mathcal{C}(F_{i-1})$. On each region $C$,*

   (a) *For $1 \leq k \leq n_0$, compute the intersections of $k$ bent hyperplanes from the new layer with $n_0 - k$ bent hyperplanes from previous layers, the latter of which are selected so that their intersection forms an $n_0 - k$-face of $C$.*

   (b) *Evaluate $F_{ij}(x)$ for each computed intersection $x$. Then keep $x$ as a vertex of $\mathcal{C}(F)$ if and only if $F_{ij}(x) = F_{ij}(C)$ for $i \leq k - 1$, following Lemma 23.*

## 4.2 Obtaining decision boundary topology from the sign sequence complex

We next produce a chain complex enabling us to compute Betti numbers of the decision boundary. Readers unfamiliar with algebraic topology may appreciate the overview by Ghrist (2014), sections 4-6. The characterization of $\mathcal{C}(F)$ as dual to a cubical complex permits us to define a straightforward mod-two cellular coboundary, which is the transpose of the boundary operation on the cubical complex.

**Lemma 21.** *The face poset of $\mathcal{C}(F)$ is the opposite poset of the face poset of $\mathcal{S}(F)$, and the (mod-two) cellular boundary map of $\mathcal{S}(F)$ is dual to the (mod-two) cellular boundary map of $\mathcal{C}(F)$.*

In other words, if $C$ is a polyhedron in $\mathcal{C}(F)$, then replacing one location for which $S(C)_{ij} = 0$ with $\pm 1$ gives the sign sequence of a polyhedron containing $C$, and doing this operation for all locations computes all terms in the cellular coboundary of $C$ (ignoring orientation). We recover the decision boundary of a network by noting that the decision boundary is the subcomplex of cells $C$ in $\mathcal{C}(F)$ whose faces $D$, including vertices, all satisfy $F(D) = 0$. By locating the vertices in $\mathcal{S}(F)$ which have a 0 as the last entry in their sign sequence, this coboundary operation, with image restricted to those cells whose last sign sequence entry is zero, gives a cochain complex of cells of the decision boundary, for example as pictured in Figure 2.

In general, the presence of unbounded cells makes this map not quite correspond to a cellular chain complex. In particular, not every edge has two vertices. By adding a single 'vertex at infinity' to unbounded edges, the corresponding cochain complex has a straightforward interpretation as the cochain complex of the one-point compactification of the decision boundary.

## 4.3 Numerical stability and algorithmic complexity

Naively, if we compute a solution $x$ to $F_{ij}(x) = 0$, and then numerically evaluate the node map $F_{ij}(x)$, the result may not be exactly zero due to floating point error. However, we show in Lemma 23 (and the subsequent discussion) that machine epsilon-level errors obtained when solving for the location of a vertex will not generally lead to errors in computing the sign sequence of a vertex. When determining the sign sequence of a vertex, which of its signs are zero is determined by which hyperplanes were intersected, and the remaining signs are stable to small perturbations, since the sets $F_{ij} > 0$ and $F_{ij} < 0$ are open sets. As long as the error in computing solutions to linear equations is small compared to the size of the cells in the polyhedral complex, the proposed algorithm will find the correct sign sequence of each vertex, and as a result the correct combinatorics of the polyhedral complex.

Furthermore, as deep ReLU networks only have polynomially many regions in the number of hidden units, at least on average at initialization (Hanin & Rolnick, 2019a), and the number of possible combinations of $k$ neurons from $n_i$ neurons together with $n_0 - k$ neurons from $n_0 + ... + n_{k-1}$ neurons is also polynomial in the total number of hidden units, it is possible to obtain the canonical polyhedral complex $\mathcal{C}(F)$ in polynomial expected time in the number of hidden units (Lemma 24). Further discussion of our algorithm's expected performance related to algorithms discussed in section 2 can be found in the appendix, in the discussion following this lemma.

## 5    APPLICATIONS TO NETWORK DECISION BOUNDARIES AT INITIALIZATION

We use this theoretical framework to make experimental observations. We obtain statistics about the decision boundaries of binary classification networks, and find stark differences in the behaviors of shallow and deeper network architectures. To our knowledge, this is the first experimental determination of the *exact* topology of a large collection of decision boundaries with input dimension greater than two while having more than one hidden layer.

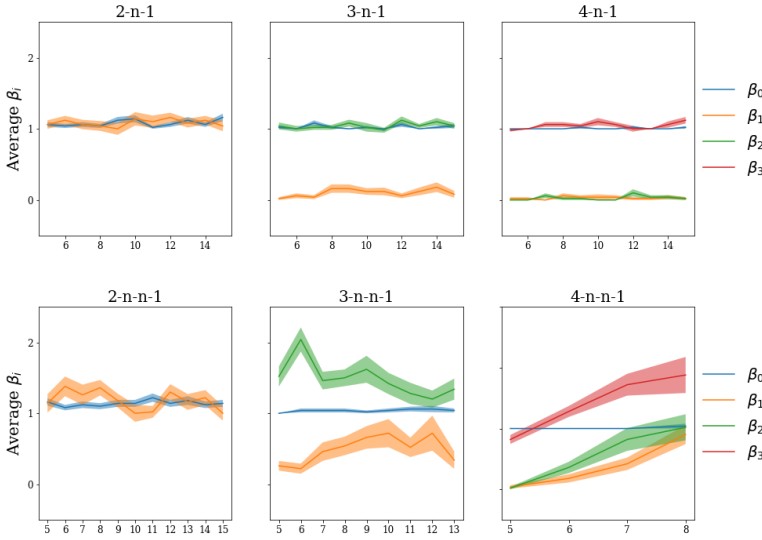

Figure 4: Average Betti numbers of the network decision boundary at initialization for a range of networks of shallow architecture (top) with deep architecture (bottom), together with standard error. The distribution of topological properties of the decision boundaries at initialization is surprisingly constant for shallow networks, and more variable for deeper networks.

### 5.1    EXPERIMENTAL DESIGN

We randomly initialize 50 networks of fully-connected architectures $(k, n, 1)$ and $(k, n, n, 1)$ for $2 \le k \le 4$ with standard normal weights and biases (See Figure 2). We will call the networks of architecture $(k, n, 1)$ "shallow" and those of architecture $(k, n, n, 1)$ "deep" for the purposes of comparison in this section. We compute the canonical polyhedral complex $\mathcal{C}(F)$ for each network using an implementation of the algorithm described in Section 4.1 using Pytorch linear algebra solver (Paszke et al., 2019). We then obtain the Betti numbers $\beta_i$ for $0 \le i \le k - 1$ of the decision boundary of the network at initialization, by constructing the boundary map determined in Lemma 21, with additional point at infinity. The Betti numbers were obtained using a Sage implementation of general chain complexes (The Sage Developers, 2018). The resulting Betti numbers provide a measure of topological complexity of the decision boundary at initialization.

### 5.2    RESULTS

We observe that the topology of the decision boundary for shallow networks, regardless of input dimension, remains constant over the range of dimensions investigated. (See Figure 4). In contrast, for deep networks, there is both greater variability in the distribution of the topology of the decision boundary, and increasing width seems to, at least for $n_0 > 2$, lead to the the topological properties of the decision boundary appearing to change in distribution as the width $n$ increases. We conjecture that a plausible explanation is that deep networks appear to require greater width before their network functions converge to Gaussian processes in distribution (de G. Matthews et al., 2018).

The Betti numbers of the decision boundary, compactified with distinguished point at infinity, measures the number of bounded and unbounded components of the decision boundary. Since all un-

bounded components are compactified by attaching them to the same point, in the compactification they correspond to $(n_0 - 1)$-cycles belonging to the same connected component, which are counted by $\beta_{n_0-1}$. So the number of bounded and unbounded components can be computed by $\beta_0 - 1$ and $\beta_{n_0-1} - \beta_0 + 1$, respectively.

We observe that *bounded* connected components of the decision boundary at initialization are rare, with frequency decreasing with input dimension in both shallow and deep networks: For example, $8.1\%$ of networks of the form $(2, n, 1)$ contain at least one bounded component, whereas only $0.05\%$ of networks of the form $(4, n, 1)$ contain as much at initialization. Furthermore, regarding the number of unbounded components, across all shallow networks investigated, the largest number of unbounded components observed at initialization was 3, with a mode of 1 (average 1.0, 1.03 and 1.04, for $n_0 = 2, 3$, and 4 respectively). In deeper networks, in contrast, the number of unbounded components at initialization appears to be on average greater $(1.1, 1.4, 1.4$, respectively) reaching maximum observed values of 5, 7 and 12 for $n_0 = 2, 3$, and 4 respectively. However, the most common observation is still that a network at initialization has one unbounded connected component, and whether there is any trend associated with width is unclear. These observations lend additional credence to the notion that depth has a stronger influence than width on the topological complexity that a network can be easily trained to express.

## 6 CONCLUSION AND FURTHER DIRECTIONS

We have presented a new combinatorial characterization of ReLU network functions, and demonstrate its usefulness for obtaining topological information about networks which was previously difficult to access. The experiments illustrate the utility of this characterization for driving further experimental research on the properties of ReLU networks in different conditions. Furthermore, we believe this framework proposed could be used to derive additional properties of ReLU networks, as theoretically, we have provided a foundation to study the structure of the canonical polyhedral complex across all dimensions. In practice, a drawback to using this algorithm in empirical work is that the algorithm is still an exponential process in the input dimension, so realistically only low-dimensional slices of the true decision boundary of a network can be investigated empirically.

We primarily believe that this tool can be useful to theoreticians by providing a local characterization of vertices of $\mathcal{C}(F)$ which can be used to obtain geometric properties of ReLU functions. For example, access to solid angles would allow for extensions to work such as Balestriero et al. (2019) on distributional properties of local curvature of decision boundaries. This local characterization additionally makes piecewise linear Morse theory as in Grigsby et al. (2022); Grunert (2017) applicable to $\mathcal{C}(F)$. Morse theory describes local extrema and saddle points by diagonalizing a second derivative matrix. Replacing a 0 with $\pm 1$ in a vertex's sign sequence identifies opposite edges incident to a vertex, giving axes for a "piecewise linear second derivative."

Another possible future application of this work is to analyze topological generalization of networks. A key indicator of a network's generalizability is whether its sublevel sets have the appropriate topological properties (Bianchini & Scarselli, 2014). In addition, the architecture of a classification network influences the topology of the expressible decision boundaries of that network (Guss & Salakhutdinov, 2018). Empirically, topological data analysis provides practical approximation for low-dimensional features of high dimensional data. While approximations exist to obtain the topological properties of a network's decision boundaries using topological data analysis (Li et al., 2020), these properties are dependent on the geometry of the network function. In places where the network's decision boundary has high curvature, the approximation methods may lead to inaccuracy between the true topology of a network's decision boundary and the topology which is approximated by persistent homology methods, but it is precisely those locations where a network is vulnerable to adversarial examples (Fawzi et al., 2018). A measure of the true topology of the decision boundary could provide a metric for comparison.

We lastly believe it is possible to obtain an explicit understanding of the evolution of a network's decision boundary through a training path. We conjecture that the changes in the facet structure of $\mathcal{C}(F)$ should be "discrete" in that they would only change at finitely many locations in a general training path, which would thus open an avenue for tracking the vertices of $\mathcal{C}(F)$ as the network trains in parameter space, and establishing theoretical limits on the possible discrete changes that may occur to the set of vertices of $\mathcal{C}(F)$ during training.

## REPRODUCIBILITY STATEMENT

We have provided source code and examples in the supplementary materials which can be used to reproduce our computational algorithms. The proofs of all numbered claims in the body of the paper may be found in the appendix.

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

## A APPENDIX

### A.1 MATHEMATICAL BACKGROUND

A summary of relevant results and definitions about polyhedral complexes and complexes arising from affine hyperplane arrangements found in Grigsby & Lindsey (2020), Sections 2-3, and Grunert (2017), Sections 1-2, with the most relevant information repeated below. Namely, we make repeated use of polyhedral geometry:

**Definition 1** (Polyhedra, Polyhedral Complex, cf. Grigsby & Lindsey (2020))**.**

- *A **polyhedron** is an intersection of the form $\bigcap_{1 \leq i \leq m} H_i^+$ for some set of (codimension 1) hyperplanes $H_1, ..., H_m \subset \mathbb{R}^n$. Here $H_i^+$ is the half-space of $\mathbb{R}^n$ consisting of the union of the hyperplane $H_i$ and one of the connected components of $\mathbb{R}^n \setminus H_i$.*

- *We will say a point is on the **interior** of a polyhedron $P$ if it is on the interior of $P$ with respect to the subspace topology of the affine span of $P$, except when $P$ is a point, in which case by convention its interior is nonempty. We use the notation $P^\circ$ to denote the interior of $P$.*

- *A **face** of a polyhedron $P$ embedded in $\mathbb{R}^n$ is a set of the form $H \cap P$, where $H$ is a codimension 1 hyperplane in $\mathbb{R}^n$ and $H \cap P$ does not contain any interior point of $P$. (The empty set is a face of any polyhedron.) A hyperplane which intersects $P$ in a nonempty face is called a **supporting hyperplane** of $P$. All other hyperplanes which intersect $P$ are called **cutting hyperplanes** of $P$. We denote the relation "$C$ is a face of $D$" with $C \leq D$.*

- *A **polyhedral complex embedded in** $\mathbb{R}^n$ is a set of polyhedra contained in $\mathbb{R}^n$ which is (a) closed under taking faces, (b) closed under intersection, in that the intersection $P_1 \cap P_2$ is the (unique) maximal common face of $P_1$ and $P_2$, which may be empty.*

Under this definition, polyhedra may not be bounded, but they are always closed. As a result, an affine hyperplane arrangement induces a natural polyhedral decomposition on its ambient space. We use the following notation for consistency with previous work, providing a notation for affine hyperplane arrangement $R^{(i)}$ associated to an affine map $A_i$.

**Definition 2** ($\pi_j$, $R^{(i)}$, cf. Grigsby & Lindsey (2020), Definition 6.7)**.** *Let $A_i : \mathbb{R}^{n_{i-1}} \to \mathbb{R}^{n_i}$ be an affine function for $1 \leq i \leq n$. Then, we define:*

- *$\pi_j$ is the linear projection onto the jth coordinate in $\mathbb{R}^{n_i}$.*

- *$R^{(i)}$ is the polyhedral complex associated to the hyperplane arrangement in $\mathbb{R}^{n_{i-1}}$, induced by the hyperplanes given by the solution set to $H_{ij} = \{x \in \mathbb{R}^n : \pi_j \circ A_i(x) = 0\}$.*

Continuing to the framework in Grigsby & Lindsey (2020), we investigate the following class of neural network functions.

**Definition 3** (Grigsby & Lindsey (2020), Definition 2.1)**.** *Let $n_0, ..., n_m \in \mathbb{N}$. A **fully-connected ReLU neural network** with **architecture** $(n_0, ..., n_m, 1)$ is a collection $\mathcal{N} = \{A_i\}$ of affine maps*

$A_i : \mathbb{R}^{n_i} \to \mathbb{R}^{n_{i+1}}$ *for* $i = 0, ..., m$. *Such a collection determines a function* $F_\mathcal{N} : \mathbb{R}^{n_0} \to \mathbb{R}$, *the* ***associated neural network map***, *given by the composite*

$$\mathbb{R}^{n_0} \xrightarrow{F_1 = ReLU \circ A_1} \mathbb{R}^{n_1} \xrightarrow{F_2 = ReLU \circ A_2} \mathbb{R}^{n_2} \xrightarrow{F_3 = ReLU \circ A_3} ... \xrightarrow{F_m = ReLU \circ A_m} \mathbb{R}^{n_m} \xrightarrow{G = A_{m+1}} \mathbb{R}^1$$

*where ReLU refers to the function* $\max\{0, x\}$ *applied pointwise (Nair & Hinton, 2010). We say that this network has* ***depth*** $m + 1$ *and* ***width*** $\max\{n_1, ..., n_m, 1\}$. *The maps* $F_k$ *are called the* $k$*th* ***layer maps***.

As a piecewise-affine linear function, $F_\mathcal{N}$, which we simplify to $F$, defines an obvious polyhedral decomposition of input space, namely into the (largest) polyhedra on which it is affine-linear. However Grigsby & Lindsey (2020) show the utility of considering not only the decomposition which $F$ itself defines, but the common refinement of decompositions by intermediate composites.

**Definition 4.** *If* $F = G \circ F_m \circ ... \circ F_1$ *is a ReLU neural network with* $F : \mathbb{R}^{n_0} \to \mathbb{R}$ , *then we denote:*

$$F_{(k)} = F_k \circ ... \circ F_1$$

*and write that this is* $F$ ***ending at the*** $k$***th layer***.

*Likewise, we denote*

$$F^{(k)} = G \circ F_m \circ ... \circ F_k$$

*and call* $F^{(k)} : \mathbb{R}^{n_{k-1}} \to \mathbb{R}$ *by* $F$ ***starting at the*** $k$***th layer***.

Thus $F = F^{(k)} \circ F_{(k-1)}$ for any $k$. A definition for the canonical polyhedral complex $\mathcal{C}(F)$ through a universal property can be given as *the common refinement of the polyhedral decomposition of input space such that all* $F_{(i)}$ *are affine linear on cells*. For implementation, we prefer a definition through explicit identification of cells, using further language from Grigsby & Lindsey (2020):

**Definition 5** ($F_{ij}$ Grigsby & Lindsey (2020), Definition 8.1 )**.** *If* $F$ *is a ReLU neural network, the* ***node map*** $F_{ij}$ *is defined by:*

$$\pi_j \circ A_i \circ F_{i-1} \circ ... \circ F_1 : \mathbb{R}^{n_0} \to \mathbb{R}$$

In particular, the locus in input space where $F_{ij} = 0$ is of particular interest, and to draw analogies to hyperplane arrangements, we use the phrase "bent hyperplane."

**Definition 6** (Grigsby & Lindsey (2020), Definition 6.1 )**.** *A* ***bent hyperplane*** *of* $\mathcal{C}(F)$ *is the preimage of* $0$ *under a node map, that is,* $F_{ij}^{-1}(0)$ *for fixed* $i, j$.

A bent hyperplane can contain polyhedral regions with codimension less than one, but this occurs with zero probability. The conditions under which the bent hyperplanes' maximal cells are always codimension 1 are listed by Grigsby & Lindsey (2020).

The original definition of the canonical polyhedral complex $\mathcal{C}(F)$ uses the notion of a "level set complex," defined by Grunert (2017). We streamline the definition, working more directly in two ways, as needed below.

**Definition 7** (Canonical Polyhedral Complex $\mathcal{C}(F)$, cf. Grigsby & Lindsey (2020), Definition 6.7)**.** *Let* $F : \mathbb{R}^{n_0} \to \mathbb{R}$ *be a ReLU neural network with* $m$ *layers. Define* $\mathcal{C}(F)$ *as follows:*

1. *(Forward Construction) Define* $\mathcal{C}(F_{(1)})$ *by* $R^{(1)}$ *(Definition 2). Then let* $\mathcal{C}(F_{(k)})$ *be defined in terms of* $\mathcal{C}(F_{(k-1)})$ *as the polyhedral complex consisting of the following cells:*

$$\mathcal{C}(F_{(k)}) = \left\{ C \cap F_{(k-1)}^{-1}(R) : C \in \mathcal{C}(F_{(k-1)}), R \in R^{(k)} \right\}$$

*Then* $\mathcal{C}(F)$ *is given by* $\mathcal{C}(F_{(m)})$.

2. *(Backwards Construction) Define $\mathcal{C}(F^{(m)})$ by $R^{(m)}$. Then $\mathcal{C}(F^{(k-1)})$ can be defined from $\mathcal{C}(F^{(k)})$ as the polyhedral complex consisting of the following cells:*

$$\mathcal{C}(F^{(k-1)}) = \left\{ R \cap F_{k-1}^{-1}(C) : R \in R^{(k-1)}, C \in \mathcal{C}(F^{(k)}) \right\}$$

*Then $\mathcal{C}(F)$ is given by $\mathcal{C}(F^{(1)})$.*

That the two are equivalent follows from the distributivity of function preimage over set intersection (or, more generally, associativity of pullbacks). That the resulting construction is indeed a polyhedral complex is discussed thoroughly by Grigsby & Lindsey (2020) and Grunert (2017). In particular, we make use of the following lemma regarding boundary relations, rewritten so as to not require additional notation.

**Lemma 8** (cf. Grunert (2017), Lemma 2.4)**.** *Let $M \subseteq \mathbb{R}^m$ and $N \subseteq \mathbb{R}^n$ be polyhedral complexes and $f : |M| \to \mathbb{R}^n$ be continuous and affine on cells of $M$. Let $\leq$ denote face relations in the respective polyhedral complexes. If $C \leq C'$ are polyhedra in $M$, and $D \leq D'$ are polyhedra in $N$, then*

$$C \cap f^{-1}(D) \leq C' \cap f^{-1}(D')$$

*is a face relation in the polyhedral complex consisting of the cells $\{C \cap F^{-1}(D) : D \in N, C \in M\}$.*

The notion of transversality on cells will be critical for the next section. Function transversality is discussed in textbooks such as Guillemin et al. (1974).

**Definition 9** (Grigsby & Lindsey (2020), Definition 4.5 )**.** *Let $X$ be a polyhedral complex of dimension $d$ in $\mathbb{R}^n$, let $f : |X| \to \mathbb{R}^r$ be a map which is smooth on all cells of $X$ and let $Z$ be a smoothly embedded submanifold (without boundary) of $\mathbb{R}^r$. We say $f$ is **transverse on cells of** $X$ to $Z$ and write $f \pitchfork_X Z$ if the restriction of $f$ to the interior $C^\circ$ of every $k$-cell $C$ of $X$ is transverse to $Z$ when $0 \leq k \leq d$*

Lastly, we will need the following notions of *generic* regarding hyperplane arrangements and neural networks, respectively.

**Definition 10** (Grigsby & Lindsey (2020) Definitions 2.7, 2.9)**.** *A hyperplane arrangement in $\mathbb{R}^n$ is called **generic** if each all sets of $k$ hyperplanes intersect in an affine space of dimension $n - k$. A neural network is called **generic** if all of its affine maps $A_i$ have generic corresponding hyperplane arrangements, $R^{(i)}$.*

Grigsby & Lindsey (2020) also established that the union of bent hyperplanes of $\mathcal{C}(F)$ form the $(n_0 - 1)$-faces of $\mathcal{C}(F)$ with probability 1. In the next section we expand on this characterization for lower-dimensional subcomplexes.

The reader is referred to Ghrist (2014) for a brief background in algebraic topology, especially the notions of homology and cohomology, chain complexes, and duality (sections 4-6).

## A.2    NEW DEFINITIONS AND PROOFS

We begin by defining a key additional condition on neural networks, which is necessary for many of the results in this section to hold.

**Definition 11.** *Let $F$ be a ReLU neural network of depth $m$. Let $\boldsymbol{F}^{(i+1)} : \mathbb{R}^{n_i} \to \mathbb{R}$ be the neural network defined by the last $m - i$ layers of $F$ as in definition 4. Suppose, for all $1 \leq i \leq n$, $F_i$ is transverse on cells of $R^{(i)}$ to the interior of all cells of $\mathcal{C}(F^{(i+1)})$. Then we call $F$ a **supertransversal** neural network.*

The condition of network supertransversality is stronger than the notion of network transversality in Grigsby & Lindsey (2020) (Definition 8.2), but it still holds on a full-measure subset of parameter space, as we show here.

**Lemma 12.** *Supertransversality is full measure in $\mathbb{R}^P$, where $P$ is the set of network parameters.*

*Proof.* First, a single-layer neural network $F^{(m)} : \mathbb{R}^{n_m} \to \mathbb{R}$ is trivially supertransversal; $\mathbb{R}$ has one cell which is already full dimension.

Next suppose that $F^{(k)}$ is supertransversal, and let $F_{k-1} : \mathbb{R}^{n_{k-1}} \to \mathbb{R}^{n_k}$ be a network layer.

Suppose it is the case that $F_{k-1}$ is nontransverse on some cell $R$ of $R^{(i-1)}$ to some cell $C$ of $\mathcal{C}(\mathbf{F}^{(k)})$. If so, it must be the case that $F_{k-1}(R) \cap C$ is nonempty and $T(F_{k-1}(R)) \oplus T(C) \neq \mathbb{R}^{n_k}$. Call $T(F_{k-1}(R)) \oplus T(C)$ by $T_{R,C}$. If $T_{R,C} \neq \mathbb{R}^{n_{k-1}}$, then it is instead a vector subspace of less than full rank, and an affine translation of $\mathcal{C}(F^{(k)})$ by any vector in $\mathbb{R}^{n_k} - T_{R,C}$ will ensure $F_{k-1}(R) \cap C$ is subsequently empty, since $F$ is affine on $R$ and $C$ contained in an affine subspace of $\mathbb{R}^{n_k}$.

Let $\delta_{R,C}$ be the minimum distance between pairs of points in $F_{k-1}(R)$ and $C$. Since these cells are closed (though not necessarily compact), this is well defined, and furthermore if $R \cap C = \emptyset$, then $\delta_{R,C} > 0$. Let $\delta = \{\min \delta_{R,C} : F_{k-1}(R) \cap C = \emptyset\}$. Then $\delta > 0$.

Since there are finitely many cells $R$ and $C$, the set

$$\mathbb{R}^{n_{k-1}} - \bigcup_{R,C} T_{R,C}$$

is generic in $\mathbb{R}^{n_{k-1}}$, where the union is taken over only those cells where $T_{R,C}$ is not full rank. An affine translation of $\mathcal{C}(F^{(k)})$ by any vector in this set with magnitude greater than 0 but less than $\delta$ yields a supertransversal network.

Since this can be performed regardless of the weights and biases of $F_k$, and an affine translation of the input space of $F^{(k)}$ does not change its supertransversality properties, the network $F_{k-1} \circ F^{(k)}$ is supertransversal on a full-measure subset of parameter space, which completes our inductive step. $\qquad\square$

The combinatorial characterization of $\mathcal{C}(F)$ is through the following combinatorial construction called *sign sequences*. The primary use of these sign sequences is to track face relations.

**Definition 13.** *Define $s : \mathcal{C}(F) \to \{-1, 0, 1\}^N$ by $s_{ij}(C) = sgn(F_{ij}(C^\circ))$. We call $s(C)$ the **sign sequence** of the cell $C$.*

This refers to the sign of the function $F_{ij}$ when restricted to the interior of $C$. To see that this is actually a function on cells, we will see that for any points $x, y$ on the interior of $C$, $\mathrm{sgn}(F_{ij}(x)) = \mathrm{sgn}(F_{ij}(y))$.

This construction is used in the theory of oriented matroids and hyperplane arrangements, cf. Aguiar & Mahajan (2017), and in particular the construction may be used to identify polyhedra in an affine hyperplane arrangement by denoting which halfspaces and hyperplanes were intersected to form that region. However, any analogous properties for ReLU networks must be proven, as many of the properties below fail to hold for arbitrary PL manifold arrangements. We must show that the construction still provides a combinatorial description of the polyhedra of the network:

**Theorem 14.** *The function $s$ is well-defined and injective on cells of $\mathcal{C}(F)$.*

*Proof.* To see that $s$ is well-defined, suppose $x_1, x_2 \in \mathbb{R}^{n_0}$ are such that $\mathrm{sgn}(F_{ij}(x_1)) \neq \mathrm{sgn}(F_{ij}(x_2))$ for some $i, j$. We wish to show that $x_1$ and $x_2$ are not in the same cell of $\mathcal{C}(F)$. However, we see that the images $F_{i-1} \circ \ldots \circ F_1(x_1)$ and $F_{i-1} \circ \ldots \circ F_1(x_2)$ cannot be in the same cell of of $R^{(i)}$ (the induced polyhedral decomposition of $\mathbb{R}^{n_{i-1}}$ by $\mathbf{A}_i$), because they differ in their location relative to the $j$th hyperplane. Thus $x_1$ and $x_2$ are in different cells of $\mathcal{C}(F_{(i)})$. As $\mathcal{C}(F)$ is a further polyhedral subdivision of $\mathcal{C}(F_{(i)})$, $x_1$ and $x_2$ are in different cells of $\mathcal{C}(F)$. So, $s$ is well defined.

Next, suppose $C_0$ and $C_1$ are cells such that $s(C_0) = s(C_1)$. Let $x_0 \in C_0$ and $x_1 \in C_1$. We wish to show $C_0 = C_1$. We proceed by induction on layers in the forward direction.

We show first that, as a base case for induction, $x_0$ and $x_1$ are in the same cell in $\mathcal{C}(F_1)$.

Indeed since $s(C_0) = s(C_1)$, $x_0$ and $x_1$ are contained in the same cell of $R^{(1)}$, following the corresponding fact for hyperplane arrangements, this immediately means that $x_0$ and $x_1$ are in the same cell of $\mathcal{C}(F_1)$.

Now suppose as an inductive hypothesis that $x_0$ and $x_1$ are in the same cell of $\mathcal{C}(F_{(k)})$. Call $y_0 = F_{(k)}(x_0)$ and $y_1 = F_{(k)}(x_1)$. Because $\mathrm{sgn}(F_{(k+1)j}(x_1)) = \mathrm{sgn}(F_{(k+1)j}(x_2))$ for all $0 \leq j \leq n_k$, this implies that $y_0$ and $y_1$ are in the same intersection of halfspaces and hyperplanes in the co-oriented hyperplane arrangement $\mathbf{A}_{k+1}$, that is, the same cell of $R^{(k+1)}$. Therefore, as $x_1$ and $x_2$ are in the same cell $C$ of $\mathcal{C}(F_{(k)})$ and their image is in the same cell $C'$ of $R^{(k+1)}$ we conclude $x_0$ and $x_1$ are in the same cell in $\mathcal{C}(F_{(k+1)})$ given by

$$C \cap (F_{(k+1)})^{-1}(C')$$

That this is a unique polyhedral cell in $\mathcal{C}(F_{(k+1)})$, follows the work in Grunert (2017), Lemma 2.5. By induction, as $F$ is composed of finitely many layers, $x_0$ and $x_1$ are in the same cell of $\mathcal{C}(F)$. $\square$

Figure 5: Even if $M_i$ are codimension-1 connected co-orientable PL manifolds embedded in $\mathbb{R}^n$ whose intersection subdivides $\mathbb{R}^n$ into polyhedral regions, and the resulting polyhedral subdivision is dual to a cubical complex, it is possible that labeling each region by its location relative to the co-orientation of those manifolds fails to be injective. This example depicts 3 PL submanifolds in $\mathbb{R}^2$ whose embedding has the aforementioned properties, but there are too many cells (14 vertices, 30 edges, and 17 2-gons) to be labeled by the 27 possible labelings in $\{-1, 0, 1\}^3$.

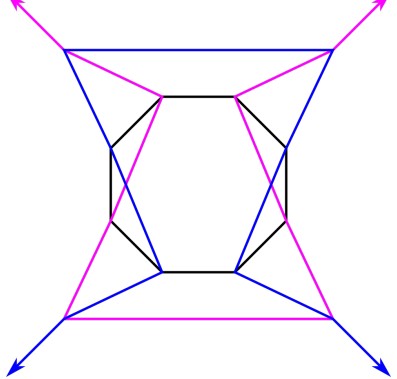

The injectivity of $s$ is special to constructions arising from hyperplane arrangements, and not general manifold arrangements. Indeed, even if a set of codimension-1 PL submanifolds subdivide $\mathbb{R}^n$ in a way which this injectivity fails; see Figure 5

Below we show that C(F) can be reconstructed from vertices and sign sequences. It is more traditional to work "top down," considering the top dimensional polytopes and their facets. But unlike theories such as hyperplane arrangements and oriented matroids, there is no guarantee that knowing the sign sequences of the top-dimensional regions allows one to deduce the sign sequences of the zero-dimensional regions (circuit-cocircuit duality does not hold). The following example is an explicit illustration of this fact.

**Theorem 15.** *There exists a pair of networks $F_1$ and $F_2$ such that the set of strings encoding the activation patterns in the interiors of the cells of $\mathcal{C}(F_1)$ and $\mathcal{C}(F_2)$ are equal, but the polyhedral complexes $\mathcal{C}(F_1)$ and $\mathcal{C}(F_2)$ are not combinatorially equivalent.*

*Proof.* We provide an example in Figure 6. Two neural networks $F_1 : \mathbb{R}^2 \to \mathbb{R}^4 \to \mathbb{R}$ and $F_2 : \mathbb{R}^2 \to \mathbb{R}^4 \to \mathbb{R}$ have the pictured canonical polyhedral complexes. One has a bounded decision boundary, whereas the other is unbounded. However, both have identical sign sequences of their top-dimensional regions, given by:

(-1, 1, -1, 1, 1),    (-1, 1, 1, -1, 1),    (1, -1, -1, -1, 1),    (-1, 1, -1, 1, -1),    (-1, -1, 1, -1, 1),
(-1, 1, 1, 1, -1),    (1, -1, -1, 1, -1),    (1, -1, 1, -1, 1),    (-1, -1, 1, 1, -1),    (-1, 1, 1, 1, 1),
(-1, -1, -1, 1, -1),    (1, -1, -1, 1, 1),    (-1, -1, 1, 1, 1),    (1, -1, 1, 1, -1),    (1, -1, 1, 1, 1),
(1, 1, -1, 1, -1),    (1, 1, -1, 1, 1)

Figure 6: The two canonical polyhedral complexes pictured below have differing combinatorics and differing topology of their decision boundaries, but the set of sign sequences of the top dimensional regions is equal (see Theorem 15). Explicit weights and biases for this construction are available in the code provided with the supplemental material.

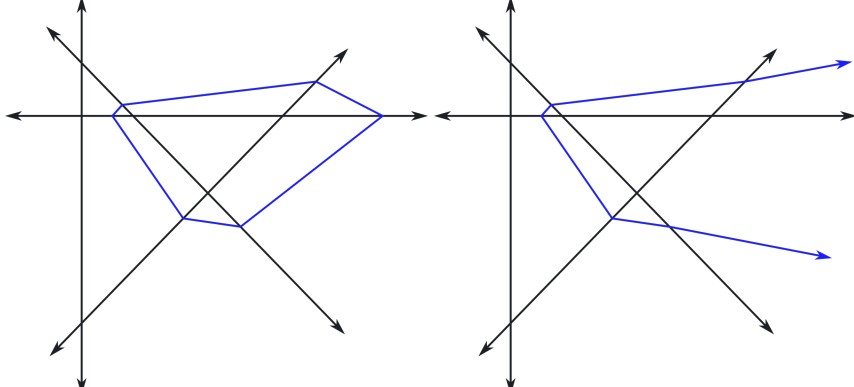

That these canonical polyhedral complexes have the same sign sequences of their top-dimensional regions is more easily seen by looking at the differences between the two pictures, which only depend on the blue "decision boundary." Each region which is subdivided into two regions by the blue bent hyperplane in the left image is also subdivided by the blue bent hyperplane in the right image.

$\square$

As illustrated in Grigsby & Lindsey (2020), it is not always the case that the preimages $F_{ij}^{-1}(0)$ are $n_0 - 1$ dimensional. In order to establish dimension, supertransversality and genericity are key.

**Lemma 16.** *Let $F$ be generic and supertransversal. Let $C$ be a $k$-cell of $\mathcal{C}(F)$. Then $s(C)$ has exactly $n_0 - k$ entries which are zero. That is, $C$ is contained in the intersection of $n_0 - k$ bent hyperplanes.*

*Proof.* This is certainly true for any neural network of the form $G \circ F_m$ satisfying the condition that $F_m$ is generic as a layer map, as the bent hyperplane arrangement is equal to the hyperplane arrangement, which is a generic hyperplane arrangement (Definition 10).

We proceed via induction, using the backwards construction of $\mathcal{C}(F)$ (Definition 7).

Suppose by way of induction $F^{(i)} = G \circ F_m \circ ... \circ F_i$, and that $\mathcal{C}(F^{(i)})$ satisfies the condition that $C \in \mathcal{C}(F^{(i)})$ is a $k$-cell if and only if $C$ is contained in exactly $n_i - k$ bent hyperplanes.

Now, suppose $F_{i-1}$ is transverse on the cells of $R^{(k-1)}$ to $C$ for all $C$ in $\mathcal{C}(\mathbf{F}^{(i)})$. Consider $\mathbf{F}^{(i-1)} = G \circ F_m \circ ... \circ F_i \circ F_{i-1}$.

Let $C'$ be a cell in $\mathcal{C}(\mathbf{F}^{(i-1)})$. Note that by definition, $C'$ is given by $F_1^{-1}(C) \cap D$ for some $C \in \mathcal{C}(\mathbf{F}^{(i)})$ and some minimal (by inclusion) cell $D$ of $R^{(i-1)}$. In particular, we may assume $C'$ is not contained in any proper face of $D$. If $D$ has codimension $\ell$, then $D$ is in the intersection of exactly $\ell$ hyperplanes in $R^{(i-1)}$ by the genericity of $F$. Because $F_1$ is transverse on cells of $R^{(i-1)}$ to $C$, codim$(C')$ in the interior of $D$ is equal to codim$(C) = k$, with total codimension $k + \ell$.

As $C$ is contained in the intersection of exactly $k$ bent hyperplanes in $\mathcal{C}(F^{(i)})$, $C'$ is contained in the intersection of the preimage of precisely those same $k$ bent hyperplanes in $\mathcal{B}(\mathbf{F}^{(i-1)})$. Furthermore $C'$ is contained in the intersection of the $\ell$ hyperplanes in $R^{(i-1)}$ which intersect to form $D$, and no additional hyperplanes as $C'$ is not contained in any proper face of $D$. Thus $C'$ is contained in the intersection of precisely $k + \ell$ bent hyperplanes in $\mathcal{C}(\mathbf{F}^{(i-1)})$ and has codimension $k + \ell$. The number of zeros in $s(C')$ must be equal to the number of bent hyperplanes it is contained in, by definition. This completes the inductive step. $\square$

In order to establish additional properties regarding the existence of cells satisfying certain relations in supertransversal networks, we rely on the following lemma.

**Lemma 17.** *Let $f : M \to \mathbb{R}^n$ be a PL map affine on cells of an embedded polyhedral complex $M \subset \mathbb{R}^m$. Let $N$ be a polyhedral complex embedded in $\mathbb{R}^n$. Suppose $f$ is transverse on cells of $M$ to the interior of all cells of $N$.*

*If $C \leq C'$ is a face relation in $M$, $D \leq D'$ is a face relation in $N$, and $f(C^\circ) \cap D$ is nonempty, then $f(C'^\circ) \cap D'^\circ$ is nonempty.*

*Proof.* First we show that $f(C^\circ) \cap D'^\circ$ is nonempty. If $D = D'$ then we are done. Otherwise, consider $f(C)$, which is the image of the polyhedron $C$ under an affine map. If the affine span $A$ of $f(C)$ does not intersect the interior of $D'$, then $A \cap D'$ is contained in a proper face $E'$ of $D'$. In this case, $T(f(C)) \oplus T(E') \neq \mathbb{R}^n$, and $f$ is not transverse on $C$ to $E'$. Therefore, $A \cap D'^\circ$ is nonempty.

Within $A$ we have $\partial(A \cap D') \subseteq A \cap \partial(D')$. As a result letting $x \in C^\circ$ and $f(x) \in D'$, every open neighborhood of $f(x)$ in $A$ must have nontrivial intersection with the interior of $D'$. Take an open neighborhood $N$ of $x$ in $C^\circ$. We note that $f : C \to A$ is a submersion (locally a surjective linear map). Thus $f(N)$ is open in $A$, and $N \cap D'^\circ$ must be nonempty, so $C^\circ \cap f^{-1}(D'^\circ)$ is nonempty.

To see that $D'^\circ \cap f(C'^\circ)$ is nonempty, take $x \in C^\circ$ with $f(x) \in D'^\circ$. If $N$ is a neighborhood of $f(x)$ in $D'^\circ$ then $f^{-1}(N)$ must be an open neighborhood of $x$ in $M$, containing $x \in C$. As $C \subset \partial C'$, $f^{-1}(N) \cap C'$ is nonempty, so $f(C'^\circ) \cap D'$ is nonempty. $\qquad \square$

Now we can define a key algebraic structure which will lead us to be able to deduce the structure of $\mathcal{C}(F)$ in general, in particular an algebraic structure which allows us to generate all sign sequences which are present from the sign sequences of the vertices. The following holds for all supertransversal networks (and does not rely on the layer maps being generic).

**Lemma 18.** *Let $F$ be a supertransversal neural network.*

*If $C$ and $D$ are two cells of $\mathcal{C}(F)$, the product $S(C) \cdot S(D)$ given by:*

$$(S(C) \cdot S(D))_{ij} = \begin{cases} S(C)_{ij} & \text{if } S(C)_{ij} \neq 0 \\ S(D)_{ij} & \text{otherwise} \end{cases}$$

*is well-defined as a product between sign sequences. That is, there exists a cell $E$ in $\mathcal{C}(F)$ such that $S(C) \cdot S(D) = S(E)$ for all such cells $C$ and $D$.*

*Furthermore, $C \leq E$, that is, $C$ is a face of $E$ or equal to $E$.*

*Thus, sign sequences of a supertransversal network form a semigroup.*

*Proof.* First, this is true for any single-layer network $F : \mathbb{R}^{n_m} \to \mathbb{R}$, since it is true for hyperplane arrangements; see Aguiar & Mahajan (2017), Section 1.4 for a treatment.

Now, suppose these properties hold for any supertransversal $k$-layer neural network and inductively, using the backwards construction of $\mathcal{C}(F)$, let $F$ be a $k + 1$-layer supertransversal neural network. Then $F^{(2)} = G \circ F_{k+1} \circ ... \circ F_2$ is a $k$-layer supertransversal network and our inductive hypothesis holds for $\mathcal{C}(F^{(2)})$. We will denote the sign sequences of cells with respect to $\mathcal{C}(F^{(2)})$ by $S^{(2)}(C)$.

Let $C$ and $D$ be cells of $\mathcal{C}(F) = \mathcal{C}(F^{(2)} \circ F_1)$. Then $C = R_1 \cap F_1^{-1}(C')$ and $D = R_2 \cap F_1^{-1}(D')$, for cells $R_1, R_2$ in $R^{(1)}$ and cells $C', D' \in \mathcal{C}(F^{(2)})$, by the definition of $\mathcal{C}(F)$. Now, by inductive hypothesis $S^{(2)}(C') \cdot S^{(2)}(D') = S^{(2)}(E')$ for some cell $E'$ in $\mathcal{C}(F^{(2)})$, and $C'$ is a face of $E'$.

Denote the sign sequences with respect to $R^{(1)}$ as $S_1$. Since $R^{(1)}$ is a polyhedral complex induced by an affine hyperplane arrangement, $S_1(R_1) \cdot S_1(R_2) = S_1(R_3)$ for $R_3$ a region in $R^{(1)}$, and $R_1$ is a face of $R_3$ or equal to $R_3$.

Let $E = R_3 \cap F_1^{-1}(E')$. We wish to show that $S(C) \cdot S(D) = S(E)$, and that $C$ is a face of $E$ or equal to it. Now, $S(C)$ is obtained by $S_1(R_1)$ concatenated with $S^{(2)}(C')$, and likewise for the other cells. Since $S_1(R_1) \cdot S_1(R_2) = S_1(R_3)$ by the corresponding hyperplane arrangement and

$S^{(2)}(C') \cdot S^{(2)}(D') = S(E')$ by inductive hypothesis, by concatenation this gives $S(C) \cdot S(D) = S(E)$.

To see that $E$ is nonempty we must note that since $F$ is supertransversal, $F_1$ is transverse on $R_1$ to $C'$, and apply Lemma 17 to obtain that $R_3^\circ \cap F_1^{-1}(E')^\circ$ is nonempty.

Lastly, we recall from Lemma 8 that $C' \leq E'$ and $R_1 \leq R_3$ implies that $(F_1^{-1}(C') \cap R_1) \leq (F_1^{-1}(E') \cap R_3)$, that is, $C \leq E$. $\qquad\square$

The following properties of the product defined above are analogous to the same properties in hyperplane arrangements.

**Lemma 19.** *For all supertransversal networks, the following relations hold for all $C$ and $D$ in $\mathcal{C}(F)$, where the relation $\leq$ denotes "is a face of":*

    *1. $C \leq D$ if and only if $S(C) \cdot S(D) = S(D)$*

    *2. $S(C) \cdot S(D) = S(D) \cdot S(C)$ if and only if there is a cell $E$ with $D \leq E$ and $C \leq E$.*

    *3. $S(C) \cdot S(D) = S(C)$ if and only if all bent hyperplanes which contain $D$ also contain $C$.*

*Proof.*

    1. We have already shown if $S(C) \cdot S(D) = S(D)$ then $C \leq D$. If $S(C) \cdot S(D) \neq S(D)$ then there is some index where $s_{ij}(C) = \pm 1$ and $s_{ij}(D) = -s_{ij}(C)$. If so, then $C$ and $D$ are sent to opposite sides of some hyperplane in some layer; this cannot occur if $C \leq D$.

    2. Immediate from the previous statement and Lemma 18.

    3. $S(C) \cdot S(D) = S(C)$ if and only if for all node maps for which $F_{ij}(C) = 0$, we also have $F_{ij}(D) = 0$. But this is true if and only if all bent hyperplanes which contain $C$ also contain $D$.

$\qquad\square$

We now assemble these ideas to present a duality between the canonical polyhedral complex of a generic, supertransversal neural network and a pure cubical complex, providing a surprising amount of new structure to the combinatorics of the canonical polyhedral complex.

**Theorem 20.** *For each generic, supertransversal neural network $F : \mathbb{R}^{n_0} \to \mathbb{R}$ with at least $n_0$ hidden units in the first layer, the image of the map $S : \mathcal{C}(F) \to \{-1, 0, 1\}^n$ uniquely defines a pure $n_0$-dimensional subcomplex of the hypercube $[-1, 1]^N$ endowed with the product CW structure. We will call this subcomplex $\mathcal{S}(F)$. In the image, the vertices in $\mathcal{S}(F)$ correspond to $n_0$-cells in $\mathcal{C}(F)$, and in general the $k$-cells of $\mathcal{S}(F)$ correspond to codimension-$k$ cells of $\mathcal{C}(F)$.*

*Proof.* Recall that cubical faces of $[-1, 1]^N$ (with its product CW structure) can be identified by sequences of $\{-1, 0, 1\}^N$.

First if $F$ has at least $n_0$ hidden units in the first layer and it is generic, then $\mathcal{C}(F)$ contains vertices as some of its cells, since the intersection of $n_0$ hyperplanes in general position in $\mathbb{R}^{n_0}$ is a point. Since $\mathcal{C}(F)$ is a connected polyhedral complex, if any of its polyhedra have vertices, then all of them do (see Grigsby et al. (2022), Corollary 5.29).

For any $C \in \mathcal{C}(F)$ there is a vertex $v \leq C$. There are $n_0$ coordinates where $S(v) = 0$ by Lemma 16. Furthermore $S(v) \cdot S(C) = S(C)$ by Lemma 18. Thus $S(C)$ is equal to $S(v)$ except those places where $S(v) = 0$. But this is equivalent to the condition that the $n_0$-cell $S(v) \in \mathcal{S}(F)$ has $S(C)$ on its boundary. So, every in the image of $\mathcal{S}(F)$ is contained in an $n_0$-cube which is also in the image of $\mathcal{S}(F)$. (There are no $n_0 + 1$-cubes in the image of $\mathcal{S}(F)$ by Lemma 16.) Thus the image of $\mathcal{S}(F)$ is "pure $n_0$-dimensional" in the sense that every cube in $\mathcal{S}(F)$ is a face of an $n_0$-cube in $\mathcal{S}(F)$.

Next we show that for a given $n_0$-cube in the image of $\mathcal{S}(F)$, all its faces are in the image of $\mathcal{S}(F)$. Our strategy is to show that there exists an edge corresponding to each possible sign sequence

incident to the corresponding vertex. Then we may apply the sign sequence multiplication in Lemma 18 to obtain all remaining faces. This is equivalent to establishing that a vertex $v$ of $\mathcal{C}(F)$ has $2n_0$ neighboring edges, each of which have a 1 or $-1$ replacing a single 0 from $S(v)$. Of course, any vertex must be incident to at least $n_0$ edges since it belongs to a polyhedral complex with domain $\mathbb{R}^{n_0}$, so we show that if there exists an edge incident to $v$ with $S_{ij}(E) = 1$ while $S_{ij}(v) = 0$, then there also exists an edge with $S_{ij}(E) = -1$ (and, by symmetry, vice versa).

Suppose that this is not the case for some $v$. Then without loss of generality there exists an earliest $(i, j)$ node map satisfying that $F_{ij}(v) = 0$ but for all edges $E$ neighboring $v$ in $\mathcal{C}(F)$, $F_{ij}(E) \geq 0$, since for each edge $E$, $S(E)$ differs from $S(v)$ only in one location. Since the edge set of $v$ is nonempty, this implies that $F_{ij}$ cannot be affine on any affine subspace of $\mathbb{R}^{n_0}$ containing $v$ unless $F_{ij} = 0$ on that subspace.

As $v$ cannot be a vertex of $\mathcal{C}(F_{(i-1)})$, since it is contained in the intersection of fewer than $n_0$ bent hyperplanes before $F_{ij}$, it is contained in the interior of a larger cell $C$ in $\mathcal{C}(F_{(i-1)})$. As a result, $F_{ij}$ is affine on the interior of $C$. But by the previous paragraph, this means that $F_{ij}(C) = 0$, and thus $F_{(i)}$ is not transverse on $C$ to a cell contained in $R^{(i)}$, and cannot be transverse on $C$ to any polyhedral subdivision (including $\mathcal{C}(F^{(i)})$). This implies that there is a layer of $F_{(i-1)}$ which fails to be transverse on cells, which is a contradiction.

So, if $v$ is a vertex of $\mathcal{C}(F)$, then for each node map $F_{ij}$ such that $F_{ij}(v) = 0$, $v$ has an incident edge with $F_{ij}(E) = 1$ and an incident edge with $F_{ij}(E) = -1$ by the same argument. We note by supertransversality that $S(E)$ must have $n_0 - 1$ entries which are zero. Also, since $v$ is incident to $E$, by Lemma 19, $S(E)$ must have the same entries as $S(V)$ except possibly where $S(v) = 0$. This means $S(E) = S(v)$ except for at the $(i, j)$ coordinate, as required. Since this occurs at all node maps for which $F_{ij}(v) = 0$, we are done. $\qquad\square$

Once the existing cells in $\mathcal{S}(F)$ have been located, we only need to establish an explicit duality. The majority of the work has already been done.

**Lemma 21.** *The face poset of $\mathcal{C}(F)$ is the opposite poset of the face poset of $\mathcal{S}(F)$, and the (mod-two) cellular boundary map of $\mathcal{S}(F)$ is dual to the (mod-two) cellular boundary map of $\mathcal{C}(F)$.*

*Proof.* In $[-1, 1]^N$, the cells consist of cubes which are uniquely defined by their center, at points given by sequences in $\{-1, 0, 1\}^N$. The dimension of each cube is given by the number of 0 entries in this sequence. The cellular boundary of this cube consists of cells one dimension lower, with a 1 or $-1$ replacing a 0 in the sign sequence, providing the (mod two) boundary in $\mathcal{S}(C)$. In $\mathcal{C}(F)$, if $s(C)$ is related to $s(D)$ by replacing one zero entry of $C$ with a 1 or $-1$, by Lemmas 19 and 16 that this is equivalent to $C \leq D$ and $dim(C) + 1 = dim(D)$, which is equivalent to $C$ being in the (mod two) cellular boundary of $D$. $\qquad\square$

Lastly, we prove that the process of computing $\mathcal{C}(F)$ can be done iteratively through layers, beginning with the first layer:

**Lemma 22.** *Let $F$ be a supertransversal, generic neural network.*

*The 0-cells of $\mathcal{C}(F_{(1)})$ are given by the solutions to*

$$\{W_\alpha x = b_\alpha : \alpha \subset [n_1] \ \& \ |\alpha| = n_0\}$$

*where $W$ is the weight matrix of the network and $\alpha$ denotes a subset of the $n_1$ vertices.*

*A vertex $v$ obtained by solving $W_\alpha x = b_\alpha$ satisfies $s_i(v) = 0$ iff $i \in \alpha$.*

*Proof.* These are the vertices of a generic, affine hyperplane arrangement. $\qquad\square$

In order to compute the vertices of $\mathcal{C}(F)$ corresponding to bent hyperplanes from further layers, we loop through regions $C$ of $\mathcal{C}(F_{(k-1)})$ and solve systems of linear equations arising from $n_0$ bent hyperplanes on that region, at least one of which corresponds to a new bent hyperplane $F_{kj}$. The following lemma guarantees that if we select these combinations of $F_{ij}$ corresponding to earlier layers from only those which intersect to form cells on the boundary of $C$, we are guaranteed to obtain all new vertices in $\mathcal{C}(F_{(k)})$.

**Lemma 23.** *Let $F$ be a generic, supertransversal neural network with at least $n_0$ hidden units in its first layer.*

*If $C$ is a cell of $\mathcal{C}(F_{k-1} \circ ... \circ F_1)$, then $F_{ij}(C)$ is affine for all $i \leq k$. Call the corresponding affine map $A_{ij} : \mathbb{R}^{n_0} \to \mathbb{R}$. Then,*

1. *All 0-cells of $\mathcal{C}(F_k \circ ... \circ F_1)$ which are contained in the closure of $C$ and which are not already in $\mathcal{C}(F_{k-1} \circ ... \circ F_1)$ are the solution to a system of $n_0$ affine equations, of which $1 \leq \ell \leq n_0$ are of the form:*

$$A_{km}(x) = 0$$

*and $0 \leq n_0 - \ell \leq n_0 - 1$ equations are of the form:*

$$A_{ij}(x) = 0; i < k$$

*Here, the $A_{ij}$ of the $n_0 - \ell$ equations from earlier layers are selected such that there exists a vertex of $C$ in the intersection of the corresponding bent hyperplanes. In other words, the remaining $n_0 - \ell$ equations describe the affine span of a face of $C$.*

2. *A solution to the system of equations described in (1) corresponds to a 0-cell of $\mathcal{C}(F_k \circ ... \circ F_1)$ contained in the closure of $C$ if and only if, for all **remaining** $(i,j)$ pairs with $i \leq k-1$, we have that $s_{ij}(v) = s_{ij}(C)$.*

*Proof.* For statement (1), suppose that $v$ is in the closure of $C$, where $C$ is a cell of $\mathcal{C}(F_{k-1} \circ ... \circ F_1)$, and $v$ is a vertex of $\mathcal{C}(F_k \circ ... \circ F_1)$. By Theorem 20, $v$ is the solution to $F_{ij}(x) = 0$ for exactly $n_0$ node maps. Since $F_{ij}|_C = A_{ij}$, then $A_{ij}(v) = 0$ for those $n_0$ node maps. If $i < k$ for all of these node maps $F_{ij}$, then in fact $v$ is a 0-cell of $\mathcal{C}(F_{k-1} \circ ... \circ F_1)$. So if $v$ is a vertex of $\mathcal{C}(F_k \circ ... \circ F_1)$ and not a vertex of $\mathcal{C}(F_{k-1} \circ ... \circ F_1)$, at least one of the $F_{ij}$ must be a node map with $i = k$. Thus, any vertex of $\mathcal{C}(F_k \circ ... \circ F_1)$ which is contained in the closure of $C$ must be a solution to a system of equations of this form. For any solution of this form to be nonempty when intersecting with the closure of $C$, the the $A_{ij}$ corresponding to this system of equations must satisfy the condition that $C \cap \bigcap \{x : A_{ij}(x) = 0\}$ is nonempty. Since none of the $A_{ij}$ from earlier layers intersect the interior of $C$, the intersection of the $A_{ij}$ are describing the linear span of a face of $C$, which must contain a vertex of $C$.

For statement (2), of course if $v$ is a solution to the system of equations described in (1) and also is in the closure of $C$, then by Lemma 19, $s_{ij}(v) = s_{ij}(C)$ when $i \leq k-1$, except for where $s_{ij}(v) = 0$, which by Theorem 20 occurs for precisely the bent hyperplanes which were intersected to obtain $s_{ij}$.

In the other direction, if $x$ is a solution to the above system of equations but is not a 0-cell of $\mathcal{C}(F)$, it must not be contained in the closure of $C$. Then $x$ is contained in the interior of some other cell of $\mathcal{C}(F_{k-1} \circ ... \circ F_1)$, call it $D$, such that $D$ is not a face of $C$ (Figure A.2). If there is some bent hyperplane corresponding to one of the remaining $(i,j)$ pairs such that $s_{ij}(D) \neq s_{ij}(C)$ then we are done. Otherwise we will see a contradiction. If $S(D) = S(C)$ except at $(i,j)$ pairs corresponding to some of the $A_{ij}$, then by our selection of equations earlier, there is a face $E$ of $C$ which has the sign sequence equal to zero at these coordinates (contained in the intersection of the solution of $A_{ij}x = 0$). If $E = D$, then $D$ is a face of $C$ and we have a contradiction. The only other option is that $E$ is a proper face of $D$ by Lemma 19. The intersection of the solutions to $A_{ij} = 0$ contains the affine span of $E$, so the intersection of these with the closure of $D$ is contained in a proper face of $D$, and so $x$, an element of this intersection, cannot be in the interior of $D$. This contradicts our assumption that $x$ is in the interior of $D$.

This shows that if $x$ is a solution to the above system of equations but is not a 0-cell of $\mathcal{C}(F)$, then there exists some $(i,j)$ pair with $i \leq k-1$ such that $s_{ij}(x) \neq s_{ij}(C)$ and which does not correspond to the hyperplanes which were intersected. $\square$

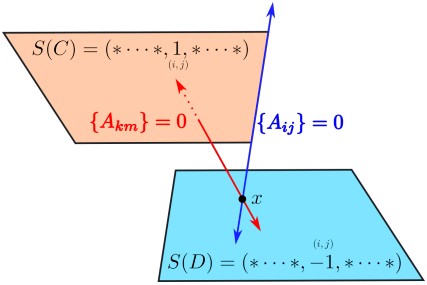

Figure 7: When determining if a solution $x$ to the system of equations in Lemma 23 is a vertex of $\mathcal{C}(F_{(k)})$, we look at its sign sequence. However, its sign relative to $\{A_{ij}\}$ is numerically unstable. We would like to guarantee it belongs to the closure of $C$ by evaluating the node maps which do not include $A_{ij}$. A concern is that it is contained in a different cell $D$ with identical signs to $C$ in $\mathcal{C}(F_{(k-1)})$ except possibly in the locations of the $A_{ij}$ which we intersected, which would make this task impossible. The argument in part (2) shows this does not occur, and the situation pictured above is impossible.

A note on numerical stability: Suppose that in step (1) we only find an approximate intersection $\tilde{x}$ instead of an exact intersection $x$, with $|\tilde{x} - x| < \varepsilon$ determined by machine precision. According to the above lemma, if we do locate the exact intersection $x$, determining whether this point $x$ is a vertex of the polyhedral complex $\mathcal{C}(F)$ relies only on determining the sign of $F_{ij}(x)$ for indices $(i, j)$ corresponding to the bent hyperplanes which were not intersected. Under the supertransversality assumptions, $F_{ij}(x)$ is strictly nonzero in these $(i, j)$ coordinates. We also note that $F_{ij}$ is continuous, so there is an open neighborhood $N$ containing $x$ where, for all $\tilde{x} \in N$, we have $F_{ij}(\tilde{x}) = F_{ij}(x)$ for all relevant $i, j$. We can therefore expect that, despite possible numerical error in computing $\tilde{x}$, the computation of the sign sequence of the corresponding vertex is stable under these perturbations, since by recording which bent hyperplanes were intersected via signs we have eliminated the unstable operation of determining which signs were 0.

We finally discuss the question of algorithmic complexity.

**Lemma 24.** *Let $F : \mathbb{R}^{n_0} \to \mathbb{R}$ be a randomly-initialized ReLU neural network satisfying the conditions in (Hanin & Rolnick, 2019a).*

*If $N = \sum_{i=1}^m n_i$ is the number of hidden units of $F$, then the average number of linear equations and sign sequence evaluations required for the computation of $\mathcal{S}(F)$ is $O(N^{2n_0+1})$.*

*Proof.* The first step in Lemma 22 involves solving $\binom{n_1}{n_0}$ equations, which is $O(n_1^{n_0})$. Then in the the recursive step in Lemma 23, in each subsequent layer $i \geq 2$, there are an average of $O((n_1 + \cdots + n_{i-1})^{n_0})$ regions to iterate through (Hanin & Rolnick, 2019a). In each region, there are fewer than $\binom{n_1 + \cdots + n_i}{n_0}$ new equations to solve. This yields the following big-O upper bound for the average computational complexity of this algorithm:

$$n_1^{n_0} + \sum_{i=1}^m (n_1 + \cdots + n_{i-1})^{n_0} n_1 + \cdots + n_i)^{n_0} \leq N^{n_0} + \sum_{i=1}^m N^{n_0}$$

Since $N = n_1 + \cdots + n_m$, and each of the $n_i \geq 1$, it must be the case that the depth $m \leq N$, so this expression is

$$O(N^{2n_0+1})$$

This is polynomial in $N$ and exponential in $n_0$. $\qquad\square$

Compare to some possible naïve approaches to computing the face relations of $\mathcal{C}(F)$. First, consider the approach of computing which regions are present and then computing their intersections. First, naïve search for all regions' activation patterns in $(-1, 1)^N$ would require determining if there

are solutions to $2^N$ linear inequalities in $N$ variables, which would then define highly redundant descriptions of polyhedral regions. This can be done more optimally than checking each set of inequalities (Serra et al., 2018; Zhang & Wu, 2020). However, without having tracked sign information, following this search, to determine if two polyhedra share a face, and find the dimension of that face, requires finding whether the union of two sets of $N$ linear inequalities is consistent, with the expectation that there will be cancellation of redundancies by the *exact equality* of some linear combinations of these linear inequalities in order to observe the existence of lower-dimensional faces. Numerical error in the expression of these linear inequalities can thus lead to catastrophic failure in identifying shared lower-dimensional faces, especially the dimension of those faces. Alternative methods of tracking which vertices are present in a way which could hypothetically allow for tracking pairwise intersection between cells, in contrast, currently track the full face poset (Yang et al., 2020; 2021), and are thus storage-intensive.

## B  ADDITIONAL EXPERIMENTAL OBSERVATIONS

Here we report additional distributional information about the topology of decision boundaries of randomly initialized ReLU networks with different architectures. Table 2 summarizes distributional information about the Betti numbers of the decision boundary, and Table 3 summarizes information about the connected components of the decision boundary. Figure 8 gives additional distributional information for selected architectures. While shallow architectures again have a very constant distribution of the number of unbounded components even across input dimension, the number of unbounded components seen at initialization in deeper architectures is much greater, and the distributional variability with width is apparent.

Table 2: Betti numbers of the compactified decision boundary dependent on architecture, across the range of widths studied (50 networks of each architecture). In $\beta_{n_0-1}$, deeper architectures exhibit greater variability and greater apparent change with width across the range of widths studied.

| | $\beta_0$ | | $\beta_{n_0-1}$ | |
|---|---|---|---|---|
| **Shallow Architectures** | | | | |
| | Average | SE | Average | SE |
| $(2,5,1)$ | 1.06 | 0.034 | 1.06 | 0.059 |
| $(2,15,1)$ | 1.16 | 0.052 | 1.04 | 0.075 |
| $(3,5,1)$ | 1.02 | 0.019 | 1.04 | 0.048 |
| $(3,15,1)$ | 1.04 | 0.027 | 1.04 | 0.040 |
| $(4,5,1)$ | 1.00 | 0.000 | 0.98 | 0.020 |
| $(4,15,1)$ | 1.02 | 0.020 | 1.12 | 0.046 |
| **Deep Architectures** | | | | |
| | Average | SE | Average | SE |
| $(2,5,5,1)$ | 1.16 | 0.05 | 1.14 | 0.13 |
| $(2,15,15,1)$ | 1.14 | 0.05 | 1.00 | 0.10 |
| $(3,5,5,1)$ | 1.00 | 0.00 | 1.52 | 0.14 |
| $(3,13,13,1)$ | 1.04 | 0.028 | 1.34 | 0.15 |
| $(4,5,5,1)$ | 1.00 | 0.00 | 0.82 | 0.07 |
| $(4,8,8,1)$ | 1.04 | 0.03 | 1.88 | 0.30 |

Table 3: Average number of bounded and unbounded components of the decision boundary dependent on architecture.

| Shallow Architectures | | | | |
|---|---|---|---|---|
| | Unbounded | | Bounded | |
| | Average | SE | Average | SE |
| $(2, 5, 1)$ | 1.00 | 0.070 | 0.06 | 0.034 |
| $(2, 15, 1)$ | 0.88 | 0.073 | 0.16 | 0.052 |
| $(3, 5, 1)$ | 1.02 | 0.053 | 0.02 | 0.020 |
| $(3, 15, 1)$ | 1.00 | 0.040 | 0.04 | 0.023 |
| $(4, 5, 1)$ | 0.98 | 0.020 | 0.00 | 0.00 |
| $(4, 15, 1)$ | 1.10 | 0.042 | 0.02 | 0.020 |

| Deep Architectures | | | | |
|---|---|---|---|---|
| | Unbounded | | Bounded | |
| | Average | SE | Average | SE |
| $(2, 5, 5, 1)$ | 0.98 | 0.140 | 0.16 | 0.052 |
| $(2, 15, 15, 1)$ | 0.86 | 0.100 | 0.14 | 0.049 |
| $(3, 5, 5, 1)$ | 1.52 | 0.142 | 0.00 | 0.000 |
| $(3, 10, 10, 1)$ | 1.38 | 0.155 | 0.04 | 0.027 |
| $(4, 5, 5, 1)$ | 0.82 | 0.0730 | 0.00 | 0.000 |
| $(4, 8, 8, 1)$ | 1.84 | 0.301 | 0.04 | 0.040 |

## C  LICENSE INFORMATION

PyTorch (Paszke et al., 2019) is under a Modified BSD license, permitting use in other projects and requiring its licensing information repackaged when its source code is redistributed. We do not redistribute its source code in our work.

Sage (The Sage Developers, 2018) is licensed under the GNU General Public License (GPL). It is free to use and distribute. We do not redistribute its source code in our work, but it is necessary to run the decision boundary topology computations.

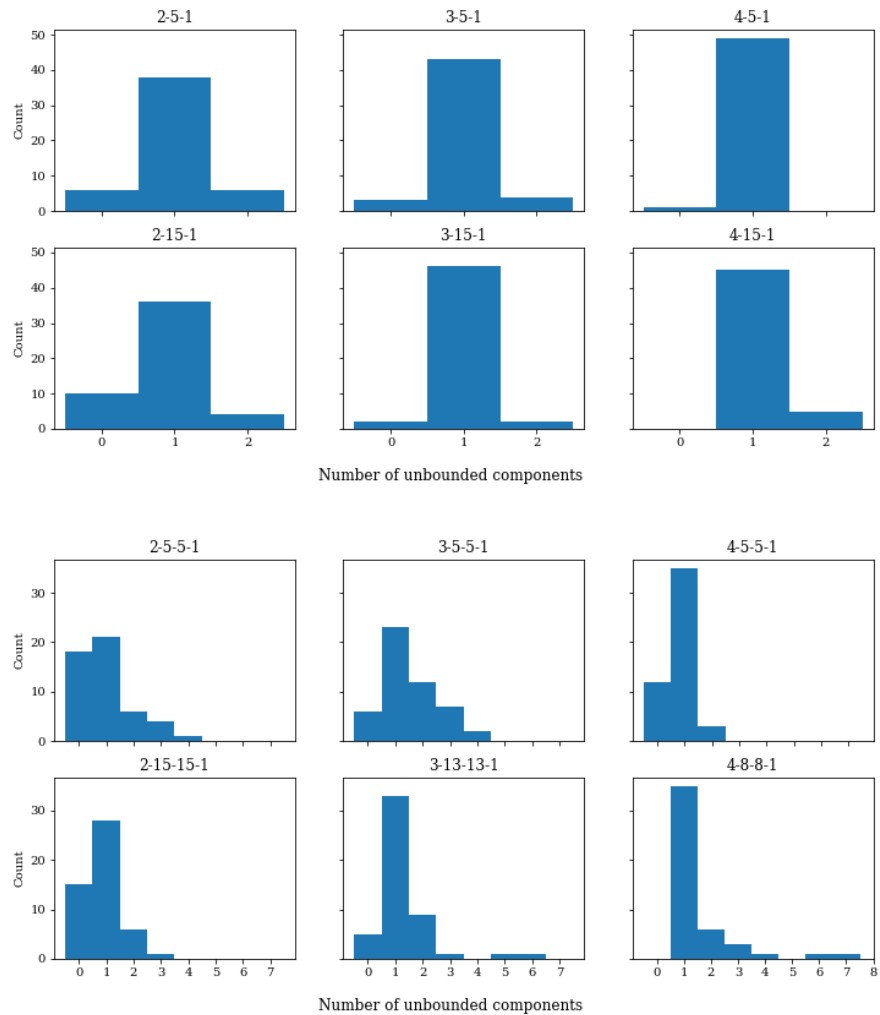

Figure 8: Distribution of the total number of connected components of the decision boundary for selected architectures.

