# OpenReview forum: "Algorithmic Determination of the Combinatorial Structure of the Linear Regions of ReLU Neural Networks"
_ICLR.cc/2023/Conference — Submitted to ICLR 2023_

### Official Review · Reviewer_Pp96 · 2022-10-24

**Confidence:** 3
**Correctness:** 4
**Technical Novelty And Significance:** 4
**Empirical Novelty And Significance:** 3
**Recommendation:** 6

**Clarity, Quality, Novelty And Reproducibility:**

Pretty well-written.

Some typos/clarifications:
-abstract: "determines" should be "determine" as it refers to locations I believe.
-theorems are numbered 14 and 20, but it would be better to have 1 and 2 so that reader is not confused. Also, the numbering is not consistent or consecutive numbers. Eg Lemma 21 appears after Lemma 23 and 22.

One More Question:
-Even though this is a nice theoretical paper, I would like to ask whether more applications can be described. Is the insight gained useful let's say for optimization purposes? In conclusion, the paper states generalization which makes sense to me.


**Strength And Weaknesses:**

Strengths are:
+ the paper builds upon the theory of fully-connected ReLU feedforward networks and yield interesting insights about the geometry of neural nets
+ the theory is then used to understand different types of networks, e.g., shallow vs deep, and to identify the exact topology of the linear regions.
+ the paper can serve as further evidence of the benefits of depth in neural nets

Weaknesses/Unclear things to clarify/Questions:
- in terms of theory, I am not convinced about the necessity of exponential dependence in the dimension. Is there an example or counterexample that shows the true dependence should behave like this? (maybe there is a simple example that I am missing out)
- in terms of experiments, I would like to have seen a comparison of the algorithm with simpler algorithms like naive searching of the input space and how it is mapped, or random sampling and tweaking of a starting point moving along some curve or sth. I know these may be trivial baselines, but it's always nice to see "failure" cases too.
-in terms of experiments, can the theory developed be used to learn something interesting about MNIST let's say? or even ImageNET? about the topological properties of the networkds used?
- I am confused about the 4th bullet in the contributions part of the intro (page 2). The statement that at initialization depth seems to be more important than the #aparams of intermediate neurons, isn't is contradicting recent work of Hanin and Rolnick that is cited in the paper (end of section 4.3)? Can there be a clarification please?


**Summary Of The Paper:**

The authors deal with the problem of identifying the different regions of space as they are partitioned by the parameters of a ReLU neural network, specifically identifying what combinatorial properties govern the induced partition.

The authors provide a combinatorial description of the canonical polyhedral complex of ReLU nets and its dual. They rely on something called the "sign sequence complex" and use it to determine the facets relations and the locations of vertices of the complex.

Having established these theoretical insights, they proceed by implementing and cheking an algorithm to obtain the sign sequence complex. The algorithm appears to work for small values of input dimension, and to be polynomial in the #neurons. Furthermore, the algorithm is numerically well-behaved.

The theory developed also is used to understand some topological properties of ReLU networks and points to the fact that depth is more important than width. (see quick question below too)

**Summary Of The Review:**

Overall, the paper looks at an interesting theoretical question regarding how the different linear regions of a ReLU network are positioned into space, how they intersect or not, what is their boundary. The main result is a characterization and an algorithm that finds the different linear regions. Then this is used to understand topological properties of small networks. I believe the algorithm is nice, and perhaps the one that is the natural algorithm to run in order to obtain the boundaries. (looping over intersections and how they are passed through the net).

---

> ### Author Response · Authors · 2022-11-11
> **Responses to Primary Questions**
>
> Thank you for your thoughtful questions. We hope we can address your primary concerns.
>
> * in terms of theory, I am not convinced about the necessity of exponential dependence in the dimension. Is there an example or counterexample that shows the true dependence should behave like this?
>
> The number of top-dimensional regions of C(F), assuming generic weights, is at least $2^{n_0}$ whenever $n_1 \geq n_0$, which is the regime required for our stated algorithm to work (by finding vertices).  This is because there are at least $2^{n_0}$ top-dimensional regions of any generic hyperplane arrangement with at least $n_0$ hyperplanes in in $R^{n_0}$. Our algorithm, as written, loops through top-dimensional regions, so there is an exponential dependence on input dimension.  We add more detail here in Lemma 24 in our current revision.
>
> * in terms of experiments, I would like to have seen a comparison of the algorithm with simpler algorithms like naive searching of the input space and how it is mapped, or random sampling and tweaking of a starting point moving along some curve  [...] can the theory developed be used to learn something interesting about MNIST let's say?
>
> A primary reason we have no other naive comparisons is that the naive approach to obtaining the exact topology is both computationally expensive and numerically unstable.
>
> A naive search for all regions’ activation patterns in (-1,1)^N would require determining if there are solutions to 2^N linear inequalities, where N is the number of intermediate neurons. This involves a search which scales exponentially, but this is not the main problem. To then determine the intersections between pairs of regions, we must compute the intersections of regions through combining these sets of linear inequalities. However, the computed rank (dimension) of such an intersection is highly numerically unstable since reducing the dimension requires exact equality of two or more of the linear equations, or possibly a linear combination of those two equations, from separate matrices which may have arisen from matrix multiplication. We attempted this approach originally, and could not find a sufficient threshold which respected both small and large regions.
>
> In our current revision, we have added aspects of this discussion to the appendix beneath Lemma 24, and refer to it in Section 4.3.
>
> Regarding searching the input space and how it is mapped, we do hope to compare the exact output of this algorithm with estimated topology from persistent homological approaches as described in our conclusion, but this requires much more experimental setup, and so this experimental comparison is beyond the scope of this paper.
>
> We do believe that our algorithm can be used to investigate the topology of the decision boundary of trained networks on real tasks such as MNIST through looking at low-dimensional slices of the decision boundary, which are computationally tractable to obtain, and we plan to investigate this in the future. We currently are working on improving the optimization of our algorithm to obtain exact decision boundaries in 10-15 input dimensions for 15-20 intermediate dimensions, which we believe will be more informative about (moderate-dimensional slices of) MNIST and similar real tasks.
>
> * I am confused about the 4th bullet in the contributions part of the intro (page 2). The statement that at initialization depth seems to be more important than the #aparams of intermediate neurons, isn't is contradicting recent work of Hanin and Rolnick that is cited in the paper (end of section 4.3)? Can there be a clarification please?
>
> While Hanin & Rolnick show that the expected number of activation patterns of ReLU networks is independent of depth (in the language of our paper, the number of top-dimensional regions of the canonical polyhedral complex), we show that the distribution of number of connected components and Betti numbers of the decision boundary of such a network experimentally does vary with depth. To see this is not a contradiction, consider that two PL functions  $\mathbb R\to \mathbb R$ can have the same number of points of nonlinearity, but have very topologically different decision boundaries. Two functions could have the same number of linear regions, but if one of these PL functions simply continues to increase, changing slopes slightly at each nonlinearity, in contrast to a second function which alternates increasing and decreasing, this will mean that any decision boundary of the first PL function will contain a single point, but the decision boundary of the second PL function could contain several points.
>
> In other words, our work effectively measures how the distribution of "critical points" apparently differs between shallow and deep networks, which is distinct from "number of regions."

---

> > ### Comment · Reviewer_Pp96 · 2022-12-05
> > **responses rebuttal**
> >
> > I appreciate your responses!

---

### Official Review · Reviewer_6J8n · 2022-11-03

**Confidence:** 4
**Correctness:** 4
**Technical Novelty And Significance:** 3
**Empirical Novelty And Significance:** 3
**Recommendation:** 6

**Clarity, Quality, Novelty And Reproducibility:**

In comparison to the version submitted to NeurIPS, I see that there were some significant changes in the second paragraph of the introduction to explain some terms that I could not understand previously. If I am not mistaken, the authors also included a third paragraph after the NeurIPS rebuttal to provide even more context, but this paragraph was not included in this submission.

I have also noticed other minor modifications elsewhere. I understand that it is difficult to start a paper with too many definitions and that it is hard to define everything that might have been used in related work, but I was hoping to see more definitions presented by section 3 rather than the reader still being referred to the appendix for definitions.



**Strength And Weaknesses:**

On the one hand, the authors are boldly approaching a rather intractable problem, and one which I believe would help us understand what neural networks encode across all their linear regions. In terms of presentation, I liked how the theorems from the appendix were brought in at the point that they would be helpful to continue the discussion. Moreover, the authors included some missing references that overlapped with their contribution and corrected some issues that I spotted when I reviewed their work previously.

On the other hand, it is still my impression that this paper expects too much from the reader and cannot be read on its own. I understand that it is hard to do justice to this work in so few pages, but I believe that would have helped to informally explain the main ideas and how they relate before going very technical (even if that means sending more of the theory to the appendix). One interesting example of that done in practice is Boris & Hanin (2019b). I was really hoping that the authors would have altered their paper to be a little more like that.

**Summary Of The Paper:**

This paper presents the mathematical theory and corresponding methods for computing a full characterization of all the linear regions of a neural network with ReLU activations. In other words, it provides the structure of the polyhedron associated with each of the linear regions defined by a rectifier network and its lower-dimensional faces, which I understand as a generalization of the face lattice that would describe a single polyhedron (i.e., one linear region). They further show that the union of the vertices across all such polyhedra along with a sign characterization of those vertices with respect to each neuron (-1, 0, +1) is sufficient to determine the facial structure of the same polyhedra. Whereas this can be combinatorially explosive, the authors argue that this can be used to evaluate the linear regions of a neural network with respect to the low-dimensional subspaces of the input.

In full disclosure, I have previously reviewed this paper when it was submitted to NeurIPS 2022. I do bid to review the same papers when I see potential in them, since I believe this would make the process easier for the authors than starting all over again. Consequently, the wording in my review is very similar to what I wrote before, but I did compare the two versions side by side to make sure my assessment remains the same.

**Summary Of The Review:**

I anticipate other reviewers objecting to the usefulness of a computationally expensive characterization such as the one studied by the authors, but I do believe this is a necessary step if we want to understand what neural networks represent.

With that said, I was hoping for deeper changes that could potentially make this paper more accessible to the machine learning community at large, and I cannot say that the current submission is at the stage yet.

---

> ### Author Response · Authors · 2022-11-11
> **Structural Modifications**
>
> Thank you for your comments and suggestions.
>
> In our current revision, the full statements of Lemmas 22-23 previously on page 6 are left for the appendix, and we add additional clarification in sections 1.1 and 3.1, including adding more figures with the goal of making the definitions in section 3.1 more clear. What we perceive as possibly the most confusing definitions in section 3 are now either stated in full (though possibly in the paragraph text) or provided with an example in Figure 1. We would appreciate your commentary on continued improvement of these sections to make this paper as clear as possible.

---

> > ### Comment · Reviewer_6J8n · 2022-11-22
> > **Following up**
> >
> > I appreciate the follow up from the authors. I see value in the paper as it is but, as I said previously, you could have a much broader reach for your work if you made it more accessible. I have pointed out to at least one example of what others have done in similar situations.

---

> > > ### Author Response · Authors · 2022-11-28
> > > **Clarifications**
> > >
> > > Thank you for your feedback. Our previous response was intended to show the places where we have aimed to make our paper more accessible in the rebuttal revision, uploaded recently during the revision period. We have looked at the example you sent, and have made new changes since the initial submission to this venue, including adding new figures and clarifying language to the main paper in sections 1.1 and 3.1 and a reorganization sending some technicalities to the appendix to make room for these clarifications. We have also added additional figures to visually clarify the meaning of several definitions and to recontextualize the purpose of the paper.

---

### Official Review · Reviewer_3fLq · 2022-11-04

**Confidence:** 4
**Correctness:** 2
**Technical Novelty And Significance:** 3
**Empirical Novelty And Significance:** 2
**Recommendation:** 5

**Clarity, Quality, Novelty And Reproducibility:**

The clarity and quality of the mathematical portion of the work is great admirable. The paper is both nicely structured and written. The only complaints are minor inconsistencies and typos outlined in "Strength And Weaknesses". Additionally, the code comes with basic documentation and clear guidance for both installation and usage. This makes the work easily reproducible. However, some of the claims that are supposed to demonstrate the advantage of the approach in the paper are unfounded or not well evidenced, see also "Strength and Weaknesses".

The reviewer is not aware of any other work discussing the computation of the canonical polyhedral complex using sign sequences.

**Strength And Weaknesses:**

Strengths
-------------

The strengths of the paper is its clarity and its novel approach. They introduce a generalization of activation patterns, which is very elegant due to its simplicity and in the sense that they induce a polyhedral complex that is dual to the canonical polyhedral complex formed by the linear regions. The authors provide code for computing the objects they introduced, and clear instructions on how to use it.

Weaknesses
-----------------

The biggest weakness of the paper is that it does not clearly answer the following question: What is the advantage of working with the sign sequence approach and the sign complex S(F) over working with the canonical polyhedral complex C(F)?
In detail:

   * There doesn't seem to be an algorithm for computing S(F) without computing C(F). Section 4.1 is titled "Obtaining the sign sequence complex", but the algorithm works by computing C(F).

   * The point about the numerical stability is unclear. Section 4.3 states: "As long as the error in computing solutions to linear equations is small compared to the size of the cells in the polyhedral complex, the proposed algorithm will find the correct sign sequence [...], and [...] the correct combinatorics of the polyhedral complex". But under the same assumptions shouldn't an approximately computed C(F) also have the same combinatorial structure as the exact C(F)?

   * The point about the algorithmic complexity is unclear. Section 4.3 states: "the number of possible combinations [...] is also polynomial in the total number of hidden units". But does that really mean that the algorithm is "polynomial expected time in the number of hidden units"? For each combination one still needs to compute the intersection points, and it is not clear whether the expected number of intersection points is polynomial in the number of hidden units?

Additionally, the paper talks about "decision boundaries", however it only considers networks with a single output dimension, i.e., decision boundaries of binary classification tasks. It would be good if the authors could comment how their work can be applied to decision boundaries of non-binary classification tasks. If the work is only restricted to decision boundaries of binary classification tasks, then this should be clearly stated.

A few very minor comments:

   * Section 3.2, first paragraph: double "a" in "may serve as a a labeling scheme"

   * Section 3.2, first paragraph: missing "the" in "encode all face relationships and [] topology of a network's canonical polyhedral complex"

   * Section 4.1, computing sign sequences: In Point 1., "F_ij" should be "F_1j"

   * Appendix A.1, Definition 1: The definition for polyhedron is wrong, it is for open polyhedra and not closed polyhedra. H_i^+ needs to be the euclidean closure of a connected component of RR^n \ H_i.

   * Appendix A.1, Definition 5: Mention that \pi_j denotes the projection onto the j-th coordinate

   * Appendix A.2, Definition 11: Some of the indices are wrong:
     - F^i maps from RR^{n_{i-1}} (by Definition 4)
     - F_i maps to RR^{n_i} (by Definition 4)
     - R^(i-1) lives in RR^{n_{i-2}} (by Definition 2)

   * Appendix A.2, Definition 13: Mention that sgn(F_ij(C)) denotes the sign of the linear function restricted to C (and not the sign of the points in the image of C under F_ij)

   * Appendix A.2, Lemma 18 onwards: sign sequences are denoted with a capital "S", and not a small "s" anymore.

**Summary Of The Paper:**

The paper studies linear regions of a ReLU neural network, which naturally form a "canonical polyhedral complex". They introduce a dual complex, the so-called "sign sequence complex", which is also a generalization of activation patterns. They prove that the sign sequences of the vertices of the canonical complex is sufficient to determine the facet poset structure of the sign sequence complex. They develop and implement algorithms for computing sign sequences and the canonical polyhedral complex, which can be used to obtain topological information of the decision boundary.

**Summary Of The Review:**

The paper describes a very interesting approach to study linear regions of ReLU networks. It is very nicely written, and even closer inspection revealed no mathematical errors in the proofs.

However, the claims that advocate for the superiority of the approach in terms of numerical stability and algorithmic are unfounded. This does not mean that the claims are wrong, just that the paper lacks proof or evidence. Additionally, the main part of the paper restricts to decision boundaries of binary classification tasks without ever mentioning it explicitly in the introduction.

---

> ### Author Response · Authors · 2022-11-11
> **Minor corrections**
>
> Thank you for your detailed commentary. In our current revision we aim to amend the typographical and indexing problems which you found as well as the points of clarity which you pointed out.
>
> We made the following changes slightly differently from your suggestions:
>
> * Section 4.1, computing sign sequences: In Point 1., "F_ij" should be "F_1j"
>
> Here we evaluate $F_{ij}$ for all (i,j) pairs and keep the sign information, but only use the (1,j) pairs to determine C(F_1).  We have clarified this in the current revision.
>
> * Appendix A.1, Definition 1: The definition for polyhedron is wrong, it is for open polyhedra and not closed polyhedra. H_i^+ needs to be the euclidean closure of a connected component of RR^n \ H_i.
>
> In our definition, we write, “Here $H_i^+$ is the half-space of $\mathbb{R}^n$ consisting of the *union* of the hyperplane $H_i$ and one of the connected components of $\mathbb{R}^n \setminus H_i$.” (emphasis added). This is equivalent to the Euclidean closure.
>
> * Appendix A.1, Definition 5: Mention that \pi_j denotes the projection onto the j-th coordinate
>
> Thank you for pointing out that this was unclear. This was included as a part of Definition 2, but we have now made Definition 2 more clearly separated into both parts.
>
> * Appendix A.2, Lemma 18 onwards: sign sequences are denoted with a capital "S", and not a small "s" anymore.
>
> Thank you for pointing this out. We will amend this, but it is not fixed yet in the current revision.

---

> ### Author Response · Authors · 2022-11-11
> **Addressing main concerns**
>
> Thank you very much for your comments and questions. We hope to address your largest concerns regarding the significance of our contributions and the support for our claims.
>
> ---
>
> * The biggest weakness of the paper is that it does not clearly answer the following question: What is the advantage of working with the sign sequence approach and the sign complex S(F) over working with the canonical polyhedral complex C(F)?
>
> Our goal is not to contrast computing C(F) versus computing S(F). Instead, we wish to demonstrate that it is possible to obtain complete information about C(F) by exploiting its duality relationship to S(F). As we now say in the paragraph at the bottom of page 2, we see S(F) as a data structure which allows us to store information about C(F) and easily derive information about the topology of C(F). At all times, understanding C(F) is our ultimate goal. To our knowledge, the previous approaches to algorithmically computing C(F) do not obtain complete information, in fact, they often only obtain information about the top-dimensional regions of C(F).
>
> To make a rough analogy, previous approaches perform a computation to determine the structure of a building and obtain a list of the bricks that were used to build it. Without the additional connecting information, you do not have enough information to determine how the building is put together. We show that by tracking sign sequence information via S(F), we can obtain the full connectivity properties of C(F),and demonstrate exactly how to do so.
>
> * There doesn't seem to be an algorithm for computing S(F) without computing C(F). Section 4.1 is titled "Obtaining the sign sequence complex", but the algorithm works by computing C(F).
>
> In our current revision, we have changed subtitles to make the point of this section more clear: we are exploiting the dual relationship between the geometric object C(F) and the combinatorial object S(F) to make encoding C(F) (particularly its full face relation structure) computationally tractable.
>
> Ultimately we are using S(F) and C(F) as dual structures which represent much of the same information in different ways, and allow for translation between two different modes of representation. In this section, observe that simply knowing the locations of the vertices of C(F_1) is not immediately equivalent to knowing all of C(F_1), since there are possibly many different polyhedral complexes which have the same set of vertices. Indeed, we must find the sign sequences of the vertices in order to know which vertices are connected to each other, which regions are present, and so on. S(F) is computed as an intermediary between locating the vertices of C(F) and having full information about what its regions are. By translating back and forth between S(F) and C(F), we can obtain *both* in a way which completely encodes all combinatorial information about the face relations of C(F).
>
> * The point about the numerical stability is unclear.  [...] under the same assumptions shouldn't an approximately computed C(F) also have the same combinatorial structure as the exact C(F)?
>
> We discuss numerical stability not to contrast computing C(F) vs. S(F), but to demonstrate that when performing our algorithm we should anticipate a correct result for the exact combinatorics of S(F) (and therefore C(F)) despite it being impossible to obtain C(F) itself exactly. Typically, the sign operation of "determining whether a function evaluated at a point is 0" requires thresholding, but when we know what hyperplanes were intersected, we have this information a priori.
>
> There are other naive approaches to obtaining this information from C(F) directly which are unstable: for example, if attempting to determine intersection structure of C(F) beginning with top-dimensional regions, intersecting two H-representations of cells in C(F) and checking that their intersection is lower-dimensional also requires setting a numerical threshold for linear equations to be equal.
>
> We have added a discussion of numerical stability claims around lemmas 23-24 in the appendix.
>
> * The point about the algorithmic complexity is unclear.... [...] the expected number of intersection points is polynomial in the number of hidden units?
>
> Thank you for pointing out this was unclear. We have added this as a lemma in our current revision (Lemma 24).
>
> * Additionally, the paper talks about "decision boundaries", however it only considers networks with a single output dimension, i.e., decision boundaries of binary classification tasks. [...] this should be clearly stated.
>
> We now clarify this point throughout our current revision. We will note here that it *is* possible to extend our work to general n-dimensional decision regions by using the *braid hyperplane arrangement* constructed from the hyperplanes $x_i = x_j$ in $\mathbb{R}^{n_{out}}$, but we do not detail this approach in our work as it is beyond the scope of our application.

---

### Decision · Program_Chairs · 2023-01-20

**Decision:**

Reject

**Justification For Why Not Higher Score:**

None of the reviewers was ready to offer a higher score for the reasons outlined above. I concur with this view.



**Justification For Why Not Lower Score:**

The article has merits as outlined above.

**Metareview: Summary, Strengths And Weaknesses:**

The paper had borderline overall final ratings 6,6,5.

* The main strengths of the paper are mathematical rigour and novelty of a proposed mathematical object to study linear regions in ReLU networks.

* The main weaknesses are insufficient evidence in support of claims about possible advantages of the proposed methods and concerns about the suitability of the work for ICLR in terms of its motivation, practical utility, and presentation.

**Summary Of Ac-Reviewer Meeting:**

There was an extended meeting between AC and all reviewers.

The meeting concluded that the paper presents a nice mathematical method, but does not sufficiently support the claimed advantages. Further, that while the paper investigates an interesting mathematical problem, it does not sufficiently motivate this in a machine learning context. While some of the points raised in the initial reviews were addressed by the authors in their rebuttal, main points of criticism persisted. The reviewers emphasised that the article has merits, but none of them was ready to offer strong support for accepting the paper at ICLR. Nonetheless, the reviewers asked me to reiterate in the meta review that the merits of the article could have weighed heavier at a different type of venue.

My conclusion after the discussion and reading the article is that it contains promising results, but that
* in case of being regarded as a theoretical contribution, it would have been stronger if it contained theoretical results beyond the most formal properties of the complexes which facilitated more direct conclusions about networks in practice.
* if regarded as a computational method or experimental contribution, the article would have been stronger if it had provided more evidence (theoretical and experimental) in support of claims on numerical stability and complexity as well as more substantive experiments illustrating application of the computation methods.